# Progressive overfilling of readily releasable pool underlies short-term facilitation at recurrent excitatory synapses in layer 2/3 of the rat prefrontal cortex

Jiwoo Shin[1,2,3], Seung Yeon Lee[2], Yujin Kim[1,2,3]*, Suk-Ho Lee[1,2,3]*

[1]Department of Brain and Cognitive Sciences, College of Natural Sciences, Seoul National University, Seoul, Republic of Korea; [2]Department of Physiology, Seoul National University College of Medicine, Seoul, Republic of Korea; [3]Neuroscience Research Institute, Seoul National University College of Medicine, Seoul, Republic of Korea

*For correspondence:
kim.y@snu.ac.kr (YK);
leesukho@snu.ac.kr (S-HL)

Competing interest: The authors declare that no competing interests exist.

## eLife Assessment

This **valuable** work explores how synaptic activity encodes information during memory tasks. All reviewers agree that the work is of very high quality and that the methodological approach is praise-worthy. The experimental data support the possibility that phospholipase diacylglycerol signaling and synaptotagmin 7 (Syt7) dynamically regulate the vesicle pool required for presynaptic release. Overall, this is a **convincing** study.

**Abstract** Short-term facilitation of recurrent excitatory synapses within the cortical network has been proposed to support persistent activity during working memory tasks, yet the underlying mechanisms remain poorly understood. We characterized short-term plasticity at the local excitatory synapses in layer 2/3 of the rat medial prefrontal cortex and studied its presynaptic mechanisms. Low-frequency stimulation induced slowly developing facilitation, whereas high-frequency stimulation initially induced strong depression followed by rapid facilitation. This non-monotonic delayed facilitation after a brief depression resulted from a high vesicular fusion probability and slow activation of $Ca^{2+}$-dependent vesicle replenishment, which led to the overfilling of release sites beyond their basal occupancy. Pharmacological and gene knockdown (KD) experiments revealed that the facilitation was mediated by phospholipase C/diacylglycerol signaling and synaptotagmin 7 (Syt7). Notably, Syt7 KD abolished facilitation and slowed the refilling rate of vesicles with high fusion probability. Furthermore, Syt7 deficiency in layer 2/3 pyramidal neurons impaired the acquisition of trace fear memory and reduced c-Fos activity. In conclusion, $Ca^{2+}$- and Syt7-dependent overfilling of release sites mediates synaptic facilitation at layer 2/3 recurrent excitatory synapses and contributes to temporal associative learning.

## Introduction

Synapses in the mammalian central nervous system undergo activity-dependent modulation of synaptic weight during repetitive stimulation, known as short-term synaptic plasticity (STP). STP is mediated by

presynaptic mechanisms that regulate the release probability ($p_r$), size of the readily releasable pool (RRP), and kinetics of vesicle replenishment of RRP (*Zucker and Regehr, 2002*; *Jackman and Regehr, 2017*; *Neher and Brose, 2018*). Short-term depression has been observed at synapses with a high baseline $p_r$, primarily because of the rapid depletion of releasable vesicles during high-frequency stimulation (HFS) (*Lin et al., 2022*). Conversely, vesicle depletion is less pronounced at synapses with a low baseline $p_r$, and cumulative increases in $p_r$ during HFS are considered to mediate short-term facilitation (STF) (*Rozov et al., 2001*; *Pan and Zucker, 2009*; *Aldahabi et al., 2024*). Central synapses exhibit a wide spectrum of STP depending on different combinations of presynaptic factors, mediating short-term depression and facilitation.

Previously, facilitation was modeled as an increase in $p_r$ after an action potential (AP) and its slow decay during the inter-spike interval (ISI), resulting in a cumulative increase in $p_r$ during HFS (*Markram et al., 1998*; *Dittman et al., 2000*). Recent studies have suggested that the number of docking (or release) sites in an active zone (AZ) is limited and only partially occupied by releasable vesicles in the resting states (*Miki et al., 2016*; *Pulido and Marty, 2018*; *Malagon et al., 2020*; *Lin et al., 2022*). Therefore, $p_r$ is determined by the vesicular fusion probability ($p_v$) and release site occupancy ($p_{occ}$). However, distinguishing the individual contributions of $p_v$ and $p_{occ}$ to $p_r$ is challenging (*Silva et al., 2021*; *Neher, 2024*). It remains also unclear whether facilitation is mediated by activity-dependent increases in the $p_v$, $p_{occ}$, or both.

Two recent models for vesicle priming and release view the 'priming' as a two-step reversible process that leads to a release-ready state. The release-ready vesicles could be vesicles in a tightly docked state (TS) or those occupying specialized docking sites (DS) and are supplied from vesicles in a loosely docked state (LS) or those occupying distinct replacement sites (RS), which are called LS/TS model (*Aldahabi et al., 2024*; *Neher, 2024*) and RS/DS model (*Miki et al., 2016*; *Silva et al., 2021*), respectively. Common to both models is the assumption that release-ready vesicles constitute only a fraction of all docked vesicles under resting conditions and that this fraction is up- and downregulated by synaptic activity.

Synaptotagmin 7 (Syt7) was recently found to play a crucial role in facilitation at different types of central synapses (*Jackman et al., 2016*; *Chen et al., 2017*; *Martinetti et al., 2022*). In addition to its role in facilitation, Syt7 accelerates refilling of the RRP after depletion and mediates asynchronous release (*Bacaj et al., 2013*; *Liu et al., 2014*; *Jackman et al., 2016*; *Tawfik et al., 2021*). Syt7-dependent synaptic facilitation is interpreted as an activity-dependent increase in $p_v$ (*Turecek et al., 2016*; *Jackman and Regehr, 2017*; *Turecek et al., 2017*; *Norman et al., 2023*). However, it remains unclear whether Syt7 contributes to facilitation through a $Ca^{2+}$-dependent increase in $p_{occ}$ (i.e. overfilling). Overfilling has been proposed as a mechanism supporting facilitation at the *Drosophila* neuromuscular junction (*Kobbersmed et al., 2020*), yet whether this process also occurs at the mammalian central synapses mediated by Syt7 has not been studied.

Each of layer 2/3 (L2/3) and layer 5 (L5) of neocortex displays intralaminar excitatory synapses between pyramidal cells (PCs), comprising a recurrent network (*Holmgren et al., 2003*; *Thomson and Lamy, 2007*). STF at recurrent excitatory synapses in a cortical network may play a crucial role in maintaining working memory (*Mongillo et al., 2008*; *Mongillo et al., 2012*; *Hansel and Mato, 2013*). Although L2/3 PCs represent task-relevant information during a delay period in working memory tasks (*Fujisawa et al., 2008*; *Ozdemir et al., 2020*), most studies on local excitatory recurrent synapses in the neocortex have focused on L5, exhibiting short-term depression (*Reyes and Sakmann, 1999*; *Hempel et al., 2000*; *Yoon et al., 2020*). Further, the synaptic properties of excitatory synapses in L2/3 remain poorly understood.

In the present study, we found that HFS of local excitatory synapses in L2/3 of the medial prefrontal cortex (mPFC) induced initial strong depression followed by delayed facilitation, whereas low-frequency stimulation induced slow monotonous facilitation. This unique form of STP could be explained by (1) high $p_v$ of RRP vesicles at rest and throughout a train stimulation; (2) the low resting vesicular occupancy of RRP and its activity-dependent increase. These conditions led to delayed facilitation through progressive overfilling of release sites during HFS with high $p_v$ vesicles beyond the resting occupancy. Syt7 knockdown (KD) in L2/3 PCs completely abolished these synaptic enhancements by inhibiting the $Ca^{2+}$-dependent increase in release site occupancy. Finally, behavioral tests for trace fear conditioning (tFC) showed that Syt7 KD impaired the acquisition of trace fear memory and reduced c-Fos expression in the mPFC. Collectively, these results support that $Ca^{2+}$- and Syt7-dependent overfilling

of release sites mediates synaptic facilitation at L2/3 recurrent excitatory synapses and contributes to temporal associative learning.

## Results

### Frequency dependence of STP at local excitatory synapses in L2/3 of the prelimbic cortex

We examined STP profiles at recurrent excitatory synapses between L2/3 PCs and at synapses from L2/3 PCs to fast-spiking interneurons (FSINs) in the prelimbic area of the rat mPFC. The PCs and FSINs were identified based on their distinct intrinsic properties and morphologies (*Figure 1—figure supplement 1*). To stimulate presynaptic PCs, L2/3 PCs were transfected with a plasmid encoding oChIEF using in utero electroporation (IUE; *Figure 1A*; *Lee et al., 2024*). Photostimulation of the oChIEF-expressing cell soma at 40 Hz consistently evoked single APs with high fidelity throughout the 600-pulse trains (*Figure 1—figure supplement 2A and B*), which was consistent with little use-dependent inactivation of oChIEF (*Lin et al., 2009*; *Hass and Glickfeld, 2016*).

Given that IUE transfected approximately 10–20% of L2/3 PCs (*Figure 1A*), we made a whole-cell patch on a non-transfected L2/3 PC or an FSIN (*Figure 1C*). To stimulate a minimal number of excitatory axon fibers, EPSCs were evoked using minimal optical stimulation (*Figure 1—figure supplement 3A*; 470 nm, 3–4 µm in radius and 3–5 ms in duration). EPSCs evoked by minimal stimulation were examined to determine the uniformity of the kinetic properties (*Figure 1—figure supplement 3B and C*). Moreover, we reassessed the kinetics of EPSCs during trains post hoc to ensure uniformity. EPSCs were evoked by 20-pulse trains at four different frequencies (5, 10, 20, and 40 Hz) to cover the range of firing rates of PCs observed in vivo during working memory tasks (*Baeg et al., 2003*). Excitatory synapses from PCs onto both other PCs and FSINs (PC-PC and PC-FSIN) showed strong (~2-fold enhancement) and lasting (several seconds) STF at 20 Hz and lower frequencies (*Figure 1D and E*). The PC-PC synapses exhibited stronger facilitation than did the PC-FSIN synapses.

Steady-state EPSCs at the PC-PC synapses were not significantly different from 5 to 20 Hz (*Figure 1F*). Frequency invariance at the PC-PC synapses suggests that $Ca^{2+}$-dependent synaptic facilitation may counteract vesicle depletion in a frequency-dependent manner (*Turecek et al., 2017*; *Lin et al., 2022*). Despite facilitation at all frequencies, the paired pulse ratio (PPR) rapidly decreased as the ISI decreased (*Figure 1G and I*). At 40 Hz stimulation, both PC-PC and PC-FSIN synapses initially underwent strong depression, then followed by facilitation in a few stimuli (*Figure 1Dd, Ed*). Such delayed facilitation after paired pulse depression (PPD) has not been previously observed in central synapses except for a slight increase in EPSC at the cerebellar synapses under artificially depolarized conditions (*Pulido and Marty, 2018*).

We confirmed that optically evoked EPSCs were strictly dependent on AP generation (*Figure 1—figure supplement 4A*), suggesting that EPSCs are evoked by optical stimulation of presynaptic axon fibers rather than by direct depolarization of presynaptic boutons. Additionally, dual patch-clamp techniques showed that the STP of optically evoked EPSCs did not differ from the electrically evoked STP of EPSCs at the PC-FSIN synapses (*Figure 1—figure supplement 4B*). Finally, we confirmed that AMPA receptor (AMPAR) desensitization was not responsible for PPD at 40 Hz, ruling out the possible effects of postsynaptic factors on STP (*Figure 1—figure supplement 5A*), and that AMPAR saturation was not significant during facilitation (*Figure 1—figure supplement 5B*).

### Delayed facilitation results from slow activation of $Ca^{2+}$-dependent vesicle replenishment at a constantly high vesicular fusion probability

The strong PPD at 40 Hz suggests a high vesicular fusion probability ($p_v$) and likely tight coupling of the releasable vesicles to a $Ca^{2+}$ source. To test this hypothesis, we examined the effects of EGTA, a slow calcium chelator, on synaptic transmission at the PC-PC synapses. The baseline EPSC amplitude ($EPSC_1$) was not affected by incubation of the slice in the bath solution containing 50 µM EGTA-AM for longer than 20 min, supporting tight coupling. Moreover, the PPR was decreased to 0.19, and subsequent facilitation slowed but was still distinct (*Figure 2A and B*). These findings indicate that the initial $p_v$ should be higher than 0.81, considering vesicle recruitment during ISI. The effects of EGTA-AM were confirmed in the same cell by measuring the baseline EPSC and PPR before and after applying

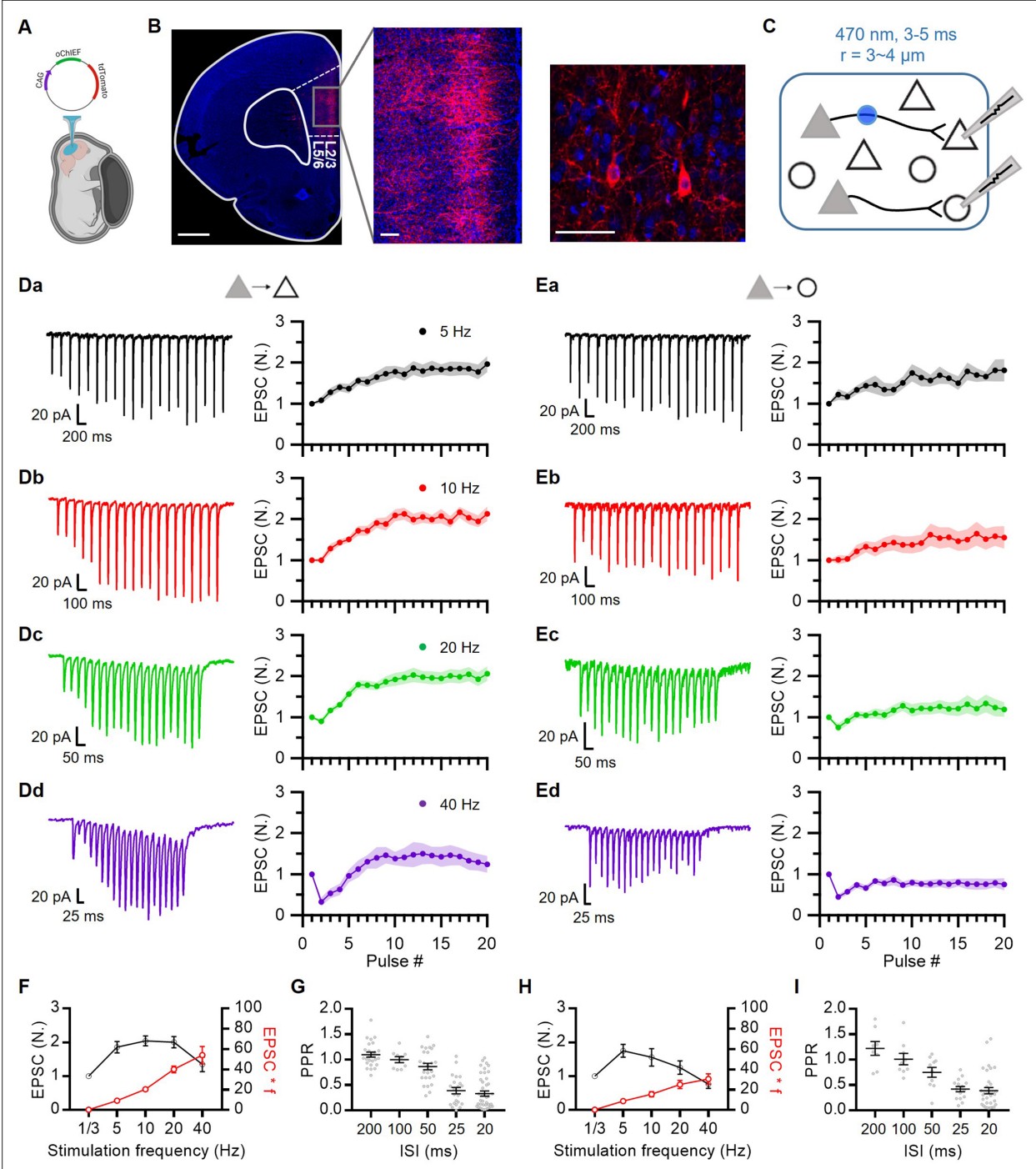

**Figure 1.** Frequency dependence of short-term synaptic plasticity at prelimbic layer 2/3 (L2/3) excitatory synapses. (**A**) In utero electroporation following injection of the plasmid (CAG-oChIEF-tdTomato) into the ventricle of an embryo (E17.5). (**B**) Representative images showing specific expression of oChIEF-tdTomato in L2/3 pyramidal cells after in utero electroporation (IUE). The red fluorescence of tdTomato clearly visualizes oChIEF-expressing cell bodies in L2/3 and axons in layer 5 (L5). Scale bar: 1 mm, 100 µm, 50 µm from left to right. (**C**) Recording schematic showing photostimulation of oChIEF-expressing axon fibers of transfected PCs (*filled triangles*) and a whole-cell recording from non-transfected PCs (*empty triangles*) or FSINs (*empty circles*). A collimated digital micromirror device (DMD)-coupled LED was used to confine the area of excitation (typically 3–4 µm in diameter) to a small region (*blue circle*) near the soma. (**D, E**) Representative traces for EPSCs averaged over 10 trials at each frequency (*left*) and average amplitudes of baseline-normalized EPSCs (*right*) during 20-pulse trains at frequencies from 5 to 40 Hz at PC-PC (D; n=12, 9, 21, 10 cells for 5, 10, 20, 40 Hz, respectively) and PC-FSIN (E; n=8, 9, 10, 10) synapses. Each data point was normalized to the average of the first EPSC. (**F**) Baseline-normalized amplitudes of steady-state EPSCs (EPSC$_{ss}$; *black symbols*) and synaptic efficacy (EPSC$_{ss}$ ×$f$; *red symbols*) as a function of stimulation frequency (f) at PC-PC synapses. EPSC$_{ss}$ was measured from the average of last five EPSCs from the 20-pulse trains (n=12, 9, 21, 10). (**G**) Paired pulse ratio (PPR) as a function of inter-spike

*Figure 1 continued on next page*

*Figure 1 continued*

intervals (n=25, 9, 25, 22, 43 for 200, 100, 50, 25, 20 ms ISI, respectively) at PC-PC synapses. (**H**) $EPSC_{ss}$ and synaptic efficacy at PC-FSIN (n=8, 9, 10, 10). (**I**) PPR at PC-FSIN (n=8, 9, 10, 15, 34). *Gray symbols*, individual data.

The online version of this article includes the following source data and figure supplement(s) for figure 1:

**Source data 1.** STP at L2/3 excitatory synapse in mPFC (*Figure 1D-E*).

**Figure supplement 1.** Intrinsic membrane properties of mPFC neurons.

**Figure supplement 1—source data 1.** F-I curve in PCs and FSINs (*Figure 1—figure supplement 1E*).

**Figure supplement 2.** Test for consistency of light stimulation in oChIEF-expressing L2/3 pyramidal cells.

**Figure supplement 3.** Properties of EPSCs evoked by minimal stimulation.

**Figure supplement 4.** Validation of STP of optically evoked EPSCs.

**Figure supplement 4—source data 1.** Comparison of STP measured by optostimulation and dual patch techniques (*Figure 1—figure supplement 4B*).

**Figure supplement 5.** Test for AMPAR desensitization and saturation.

**Figure supplement 5—source data 1.** Effects of kynurenate on STP (*Figure 1—figure supplement 5B*).

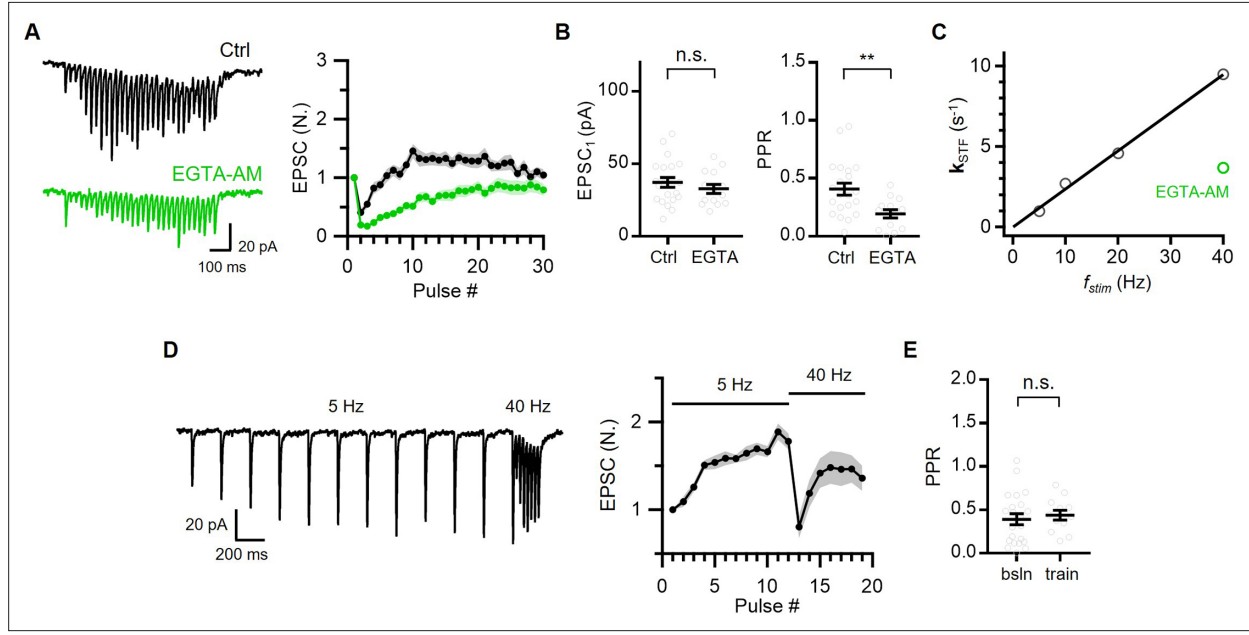

**Figure 2.** Delayed facilitation results from slow activation of $Ca^{2+}$-dependent vesicle replenishment at a constantly high vesicular fusion probability. (**A**) Representative traces (*left*) and mean baseline-normalized amplitudes (*right*) of EPSCs evoked by 30-pulse trains at 40 Hz in control (n=21; *black*) and in the presence of 50 µM EGTA-AM (n=14; *green*). (**B**) Mean values for the first EPSC amplitude ($EPSC_1$, *left*) and paired pulse ratio (PPR) (*right*) from the experiments displayed in (**A**). *Gray symbols*, individual data. (**C**) Plot of rate constants for short-term facilitation ($k_{STF}$) as a function of stimulation frequency ($f_{stim}$), showing a linear relationship. The linear regression line (*black*) is shown fitted to $k_{STF}$ values, estimated from *Figure 1*. (**D**) Representative EPSC traces (*left*) and average of baseline-normalized EPSCs (*right*) evoked by 12-pulse stimulation at 5 Hz, followed by 40 Hz 7-pulse train (n=12). Note that slowly developing facilitation was converted to rapid facilitation after strong paired pulse depression (PPD). (**E**) Mean values for PPR at 40 Hz. The baseline PPR was reproduced from *Figure 1G* and the PPR during 5 Hz train was calculated as (13th EPSC)/(12th EPSC). *Gray symbols*, individual data. All statistical data are represented as mean ± SEM; n.s.=not significant; **, p<0.01; unpaired t-test.

The online version of this article includes the following source data and figure supplement(s) for figure 2:

**Source data 1.** Effects of EGTA-AM on baseline EPSCs, PPR and STF.

**Source data 2.** Frequency-dependent acceleration of STF.

**Figure supplement 1.** Effects of EGTA-AM on EPSCs measured in the same cell.

**Figure supplement 2.** Presynaptic calcium kinetics at boutons of layer 2/3 pyramidal cells.

EGTA-AM to the bath. $EPSC_1$ remained unchanged, while PPR was reduced by half (*Figure 2—figure supplement 1*).

Notably, the application of EGTA-AM markedly slowed facilitation at 40 Hz (*Figure 2A and C*). The frequency invariance of steady-state EPSCs (*Figure 1F*) and the slower facilitation at 40 Hz in the presence of EGTA (*Figure 2A*) imply a $Ca^{2+}$-dependent increase in the refilling rate of the RRP and/or an increase in the $p_v$ of reluctant vesicles in light of previous studies (*Hosoi et al., 2007*; *Liu et al., 2014*; *Ritzau-Jost et al., 2014*; *Turecek et al., 2016*; *Ritzau-Jost et al., 2018*; *Kusick et al., 2020*). Moreover, facilitation was accelerated at higher stimulation frequencies. The plot of the rate constant for the increase in EPSCs (denoted as $k_{STF}$, the rate constant for the development of STF) as a function of the stimulation frequency ($f_{stim}$) revealed a linear relationship (*Figure 2C*). $k_{STF}$ in the presence of EGTA-AM was distinctly below this relationship, supporting the $Ca^{2+}$-dependent acceleration of facilitation. $k_{STF}$ was also accelerated when the stimulation frequency was increased from 5 to 40 Hz (*Figure 2D*; $1.27\pm0.18$/s to $24.27\pm3.44$/s). Notably, the second 40 Hz train stimulation led to strong PPD, followed by accelerated facilitation (*Figure 2D and E*). The strong PPD suggests that the vesicular fusion probability kept high during the 5 Hz stimulation, arguing against the possibility for an increase in $p_v$. Both two-step sequential priming models, LS/TS and RS/DS models, are suitable to interpret our data and we will refer to the different populations of vesicles as TS and LS vesicles, respectively (*Taschenberger et al., 2016*; *Neher and Brose, 2018*; *Lin et al., 2022*; *Neher, 2024*). Within this framework, vesicles are released from TS with high $p_v$, both in the resting state and during facilitation, and the delayed facilitation is seen as a result of $Ca^{2+}$-dependent progressive increase in the occupancy of TS vesicles on DS.

## Presynaptic calcium measurements at axonal boutons of L2/3 PCs

To test the contribution of $Ca^{2+}$ channel regulation to STP (*Dobrunz et al., 1997*; *Jackman and Regehr, 2017*), we estimated the total amount of $Ca^{2+}$ increments during 40 Hz AP trains at axonal boutons of L2/3 PCs in the mPFC. To this end, we measured AP-evoked $Ca^{2+}$ transients (AP-CaTs) using two-dye ratiometry techniques and estimated calcium binding ratio from AP-CaTs (*Figure 2—figure supplement 2A–C*). Measuring the amplitudes and decay time constants of AP-CaTs at different concentrations of Fluo-5F (150, 250, or 500 µM), and plotting them as a function of calcium binding ratio of Fluo-5F ($\kappa_B$), we estimated the amplitude of AP-CaTs ($A_{Ca} = 1.16$ µM), $Ca^{2+}$ decay time constant ($\tau_c = 43$ ms) in the absence of exogenous $Ca^{2+}$ buffer, and endogenous static $Ca^{2+}$ binding ratio ($\kappa_S = 96.4$ or $77.3$ estimated from *Figure 2—figure supplement 2D and E*, respectively; see Materials and methods). The resting $[Ca^{2+}]$ ($[Ca^{2+}]_{rest}$) was measured as 50 nM (*Figure 2—figure supplement 2F*). Estimating the total $Ca^{2+}$ increments for the first four AP-CaTs in a 40 Hz train according to *Equation 2* (Materials and methods), we found no evidence for inactivation or facilitation of calcium influx during a 40 Hz train (*Figure 2—figure supplement 2G–I*), arguing against the possibility that $Ca^{2+}$ channel inactivation and facilitation contribute to strong PPD and delayed facilitation during a 40 Hz train.

## Release sites have low baseline occupancy, and these are increased during facilitation and post-tetanic augmentation

Given that $p_v$ was greater than 0.8, the twofold increase in EPSC during 20-pulse train stimulations could not be explained by an increase in $p_v$. Recent studies have shown that vesicle release sites (N) are limited in number and are partially occupied at rest (*Miki et al., 2016*; *Pulido and Marty, 2018*; *Malagon et al., 2020*; *Lin et al., 2022*). Under such a high $p_v$, twofold synaptic facilitation should be attributed to an increase in $p_{occ}$. The vesicle replenishment rate is accelerated during HFS in a $Ca^{2+}$-dependent manner (*Figure 2A–C*; *Hosoi et al., 2007*). This may lead to the overfilling of release sites beyond their basal vesicular occupancy, which in turn may increase EPSC under high $p_v$. To test this hypothesis, we determined the quantal parameters at the L2/3 local excitatory synapses. We first measured the quantal size from asynchronous release events after replacing extracellular $Ca^{2+}$ with $Sr^{2+}$, as previously described (*Yoon et al., 2020*; *Figure 3—figure supplement 1A and B*). The mean and coefficient of variance of the quantal sizes were 24 pA and 0.25 at PC-PC synapses, respectively, and were 30 pA and 0.31 at PC-FSIN synapses (*Figure 3—figure supplement 1C*).

Next, we performed a variance-mean (V-M) analysis of phasic EPSCs at excitatory synapses during train stimulation at 5 and 40 Hz (*Scheuss et al., 2002*). Applying multiple-probability fluctuation analysis (MPFA) using intra-site quantal variability of 0.2 (*Valera et al., 2012*), the V-M plot was best fitted

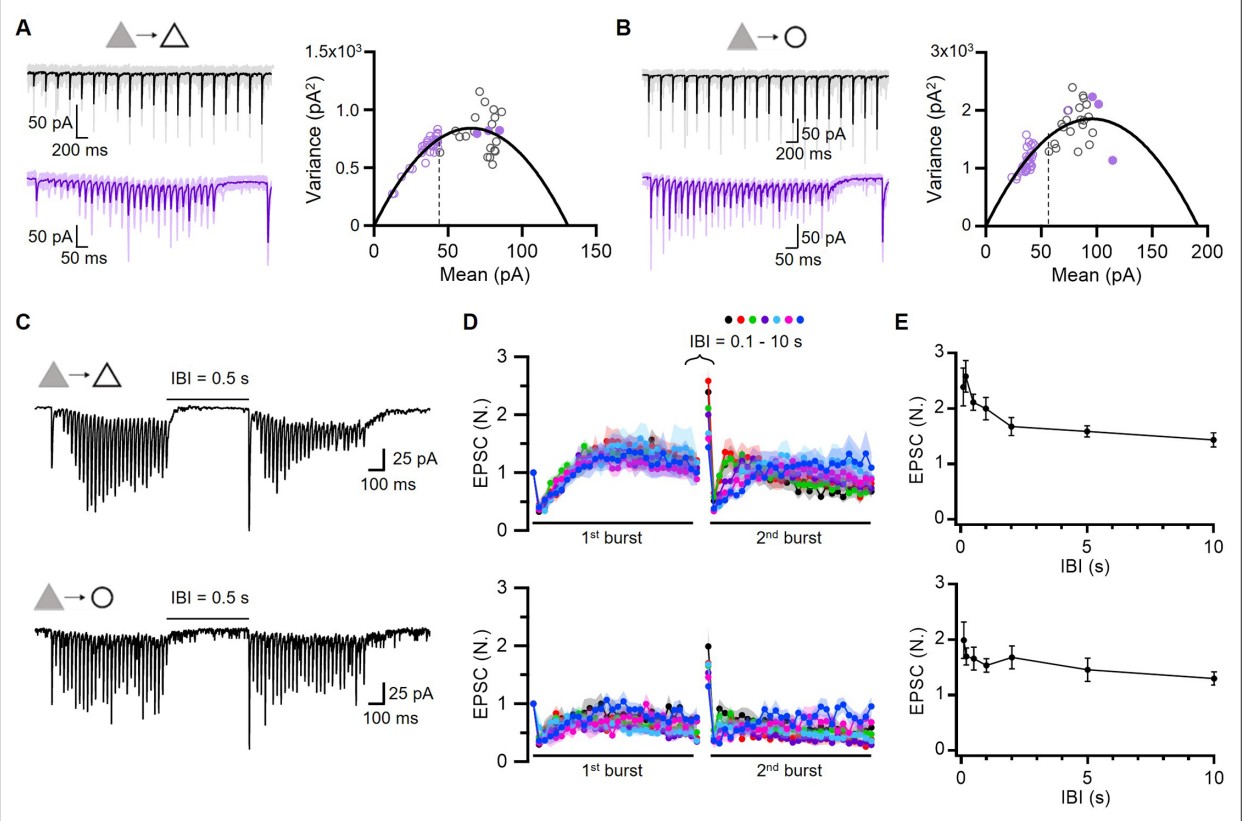

**Figure 3.** Low baseline occupancy of release sites and its increase during facilitation and post-tetanic augmentation. (**A, B**) *Left*, Representative EPSCs evoked by 5 and 40 Hz train stimulation (*black*, 5 Hz; *purple*, 40 Hz). *Right*, Variance-mean plots of EPSCs amplitude from averaged EPSCs recorded at PC-PC (A; n=12, 17 for 5 and 40 Hz, respectively) and PC-FSIN (B; n=8, 10) synapses. The data were fitted using multiple-probability fluctuation analysis (MPFA). Error bars are omitted for clarity. The 1st EPSC of the 5 Hz train (*broken line*) was used to estimate the resting level of $p_{occ}$. *Filled circles* were measured from post-tetanic augmented EPSCs (A, n=12, 12, 9; B, n=7, 7, 7 for 0.1, 0.2, 0.5 s inter-burst intervals [IBIs], respectively). (**C–E**) Post-tetanic augmentation (PTA) experiments at PC-PC (*top*) and PC-FSIN (*bottom*) synapses. (**C**) Representative traces for EPSCs evoked by double 40 Hz train stimulations separated by 0.5 s. (**D**) Mean baseline-normalized amplitudes of EPSCs evoked by double 40 Hz trains at different IBIs (0.1, 0.2, 0.5, 1, 2, 5, 10 s). *Upper*, PC-PC synapse (n=12, 12, 21, 11, 11, 16, 11 from short to long IBIs, respectively). *Lower*, PC-FSIN synapse (n=10, 9, 8, 9, 9, 7, 9). (**E**) PTA time course. The baseline-normalized amplitudes of 1st EPSC from the 2nd train were plotted as a function of IBIs.

The online version of this article includes the following source data and figure supplement(s) for figure 3:

**Source data 1.** Post-tetanic augmentation at WT synapses.

**Source data 2.** Variance-mean plots of EPSCs at WT synapses.

**Figure supplement 1.** Quantal size estimated from $Sr^{2+}$-induced asynchronous release.

**Figure supplement 1—source data 1.** Quantal sizes at PC-PC and PC-IN synapses using $Sr^{2+}$ techniques.

**Figure supplement 2.** Analysis of double failure rates supports high pv at excitatory synapses.

**Figure supplement 2—source data 1.** Matlab codes for synaptic failure analyses shown in *Figure 3—figure supplement 2*.

**Figure supplement 3.** Minimal EPSC increase upon elevated $[Ca^{2+}]_o$ indicates saturated $p_v$ at baseline.

**Figure supplement 3—source data 1.** Effects of high external $[Ca^{2+}]$ on baseline EPSCs with and without EGTA-AM.

by the number of release sites (N) of 5.3 and the quantal size (q) of 25 pA at the PC-PC synapses (*Figure 3A*). The $p_r$ for $EPSC_1$ was calculated as 0.32. We performed the same analysis for the PC-FSIN synapses and found N=5.3, q=35 pA, and $p_r$ for $EPSC_1$=0.31 (*Figure 3B*). Our estimate for N was smaller but comparable to the estimate (6.9) at excitatory synapses in L2/3 of the mouse barrel cortex (*Holler et al., 2021*).

The $p_r$ estimate was clearly smaller than the possible minimum value for $p_v$ (0.81) estimated from the PPR data (*Figure 2B*). Given that $p_r$ is a product of $p_{occ}$ and $p_v$ (*Neher, 2024*), a $p_r$ smaller than $p_v$ indicates the partial occupancy of release sites in the resting state. If synaptic facilitation were mediated by the recruitment of new release sites, it would be accompanied by an increase in the variance

of augmented EPSCs (*Valera et al., 2012*; *Kobbersmed et al., 2020*). However, the augmented EPSCs during facilitation followed the same parabola in the V-M plot (*Figure 3A and B*). Therefore, facilitation appears to be mediated by the progressive overfilling of a finite number of release sites beyond the basal occupancy. In the next section, we show that $p_v$ is close to unity based on high double failure rate upon paired pulse stimulation (*Figure 3—figure supplement 2*) and little effect of extracellular $[Ca^{2+}]$ ($[Ca^{2+}]_o$) elevation on baseline EPSC amplitudes (*Figure 3—figure supplement 3*). Assuming this, the baseline $p_r$ at the first pulse estimated from the fitted parabolic curve can be largely regarded as a baseline $p_{occ}$ (ca. 30%), and the twofold increase in EPSCs during facilitation can be attributed to an increase of up to 60% in $p_{occ}$.

We investigated the time course of recovery from facilitation and post-tetanic augmentation (PTA) by applying dual-train stimulations (30 pulses at 40 Hz) separated by different inter-burst intervals (IBIs; *Figure 3C and D*). The initial EPSCs in the second train were enhanced by more than twofold and followed by a strong PPD, suggesting that fusion probability stays high in the IBIs and thus synaptic transmission is carried by TS vesicles. The decay time course of PTA was characterized by two phases (initially fast and later slow; *Figure 3E*), which may reflect a two-phase $Ca^{2+}$ decay or two distinct recovery processes. The variance of potentiated EPSCs measured at IBIs of 0.1, 0.2, and 0.5 s was not different from that of EPSCs at the peak of facilitation (filled circles, *Figure 3A and B*). Given that the same RRP mediates both facilitation and augmentation and that post-tetanic EPSCs are carried by TS vesicles similar to the baseline EPSCs, this plot suggests that PTA, similar to STF, is mediated by an increase in the TS vesicle occupancy.

## High double failure rate and little effect of $[Ca^{2+}]_o$ elevation on baseline EPSC amplitudes suggest that vesicular fusion probability is close to one

The vesicle dynamics of STP has been explained using a simple refilling model (*Hosoi et al., 2007*; *Neher and Sakaba, 2008*). This model assumes that the vesicles are reversibly docked to a limited number of release sites (N) with forward and reverse rate constants (denoted as $k_1$ and $b_1$, respectively). Under the framework of the simple refilling model (see Materials and methods), the rate constant for the refilling of the RRP after depletion by the first pulse is the sum of the baseline $k_1$ and $b_1$. This is estimated as 23/s from the plot of PPR as a function of ISIs (*Figure 3—figure supplement 2A*). From the baseline occupancy of 0.3 and $p_{occ} = k_1/(k_1 + b_1)$, we estimated $k_1$ to be 6.9/s.

When paired APs were applied with an ISI of 20 ms, the failure rate at the 1st pulse, $P(F_1)$, was 10.6%, and the probability of two consecutive failures, $P(F_1, F_2)$, was 6.2% (*Figure 3—figure supplement 2A*). From the equation $P(F_1) = (1 - p_r)^N$, $p_r$ was calculated as 0.312 when N=6, comparable to $p_r$ (=0.32) estimated from the V-M analysis (*Figure 3A*). We calculated $P(F_1, F_2)$ under the framework of a simple refilling model, with $p_v$ and $k_1$ set as free parameters and other parameters set according to the following relationships: $P(F_1) = (1 - p_r)^N$, $p_r = p_v \cdot p_{occ}$, and $p_{occ} = k_1/(k_1 + b_1)$. The calculated values for $P(F_1, F_2)$ and their difference from the observed value (6.2%) are shown in the plane of $k_1$ vs. $p_v$ in *Figure 3—figure supplement 2B and C*. Given that $k_1$ was greater than 5/s, the difference between the calculated and observed values of $P(F_1,F_2)$ was minimal when $p_v = 1$ (*Figure 3—figure supplement 2C*). Intuitively, under a classical view ($p_v < 1$), $P_{01}$ should be larger than $P_{00}$ (subscripts 1 and 0 denote success and failure, respectively), because residual vesicles remaining after the 1st pulse and residual calcium-dependent increase in $p_v$ would reduce the 2nd failure rate. Contrary to this prediction, we observed that $P_{01} < P_{00}$.

To find optimal values for $k_1$ and $p_v$ that best explain observed probabilities shown in *Figure 3—figure supplement 2Ac*, a cost function was implemented as the sum of squared errors between observed and predicted probability values for the four combinations of success and failure of 1st and 2nd EPSCs as described in Materials and Methods. The minimum of the cost function ($5.63 \times 10^{-5}$) was found at $k_1$=5.21/s and $p_v = 0.999$, which predicted $P_{11}$=41.5%, $P_{10}$=47.9%, $P_{01}$=4.93%, and $P_{00}$=5.67%.

To further validate that baseline $p_v$ is near to unity, we examined EPSC changes following elevation of $[Ca^{2+}]_o$ from 1.3 to 2.5 mM. According to the fourth-power relationship between synaptic responses and $[Ca^{2+}]_o$ (Equation 3 in *Dodge and Rahamimoff, 1967*), a 3.24-fold increase in the EPSC amplitude was expected. However, the EPSC amplitude increased only 1.23-fold on average, a change that was not statistically significant (*Figure 3—figure supplement 3A and B*). This small response suggests that $p_v$ is already saturated at rest, limiting the dynamic range for further enhancement

through increased calcium influx. Recent morphological and functional studies revealed that elevation of $[Ca^{2+}]_o$ induces an increase in the number of TS or docked vesicles to a similar extent as our observation (*Kusick et al., 2020*; *Lin et al., 2025*), raising a possibility that an increase in $p_{occ}$ is responsible for the 1.23-fold increase in EPSC at high $[Ca^{2+}]_o$. A slight but significant reduction in PPR was observed under high $[Ca^{2+}]_o$ too. An increase in $p_{occ}$ is thought to be associated with that in the baseline vesicle refilling rate. While PPR is always reduced by an increase in $p_v$, the effects of refilling rate to PPR are complicated. For example, PPR can be reduced by both a decrease (*Figure 2—figure supplement 1*) and an increase (*Lin et al., 2025*) in the refilling rate induced by EGTA-AM and PDBu, respectively. Thus, the slight reduction in PPR is not contradictory to the possible contribution of $p_{occ}$ to the high $[Ca^{2+}]_o$ effects. Notably, when slices were pre-incubated with 50 µM EGTA-AM, elevating extracellular $[Ca^{2+}]$ from 1.3 to 2.5 mM produced no significant change in either baseline EPSC amplitude or PPR (*Figure 3—figure supplement 3C and D*), supporting that the modest $Ca^{2+}$ dependence of baseline EPSCs and PPR in the absence of EGTA is primarily mediated by a change in $p_{occ}$ rather than $p_v$.

## Pharmacological manipulations reveal specific molecular identities underlying vesicle loading processes

Our results suggest that the baseline occupancy of TS vesicles is low and that the activity-dependent progressive overfilling of DS with TS vesicles is responsible for STF and augmentation. To elucidate the specific molecular link between synaptic enhancement and $Ca^{2+}$-dependent overfilling, we examined the effects of several drugs known to affect vesicle dynamics. We first examined the roles of phospholipase C (PLC) and diacylglycerol (DAG) by applying 5 µM U73122, a PLC inhibitor, or 20 µM 1-oleoyl-2-acetyl-*sn*-glycerol (OAG), a DAG analog. The STP at 40 Hz and PTA at 0.5 s interval were examined before and after applying each drug to the bath. For both drugs, the facilitation and augmentation were reduced (*Figure 4A and B*). U73122 did not affect the basal EPSC amplitude ($EPSC_1$) but resulted in a stronger PPD, slower facilitation, and lower augmentation (*Figure 4A*). These effects of U73122 suggest the involvement of $Ca^{2+}$-induced PLC activation in progressive overfilling. In contrast, OAG increased $EPSC_1$ while maintaining a pronounced PPD (*Figure 4B*), indicating an increase in the TS vesicle pool size. Meanwhile, OAG slowed $k_{STF}$ and reduced augmentation, indicating occlusion of activity-dependent synaptic facilitation. Phorbol esters and DAG are thought to accelerate priming through Munc13 activation (*Rhee et al., 2002*; *Taschenberger et al., 2016*; *Aldahabi et al., 2022*); they also shorten the bridges that interlink docked vesicles with the AZ membrane (*Papantoniou et al., 2023*). Collectively, these suggest that PLC- and DAG-mediated signaling plays a role in the overfilling of DS underlying STF and augmentation.

Although the acceleration of $Ca^{2+}$-dependent vesicle refilling supports rapid facilitation at 40 Hz, the later part of the delayed facilitation underwent a slight depression, which was more pronounced during the second burst (*Figure 3D* and *Figure 4*). Sustained synaptic transmission is limited by the spatial availability of DS because it is undermined by accumulation of exocytic remainings in the presynaptic AZ during HFS (*Neher, 2010*; *Haucke et al., 2011*). To verify this, we examined the effects of dynasore (100 µM), a specific inhibitor of endocytosis (*Macia et al., 2006*). The treatment with dynasore did not influence $EPSC_1$ or the early phase of facilitation, but it exacerbated depression during the later phase and attenuated PTA (*Figure 4C*). This suggests that late synaptic transmission during the 40 Hz train is limited by site clearance.

Given the involvement of actin polymerization in the multiple steps of vesicle docking and recycling, we examined the effects of 20 µM latrunculin B (LatB), an inhibitor of actin polymerization. The effects of LatB were complicated: (1) $EPSC_1$ was decreased, whereas PPR was increased (*Figure 4Da*) and (2) facilitation and augmentation were reduced (*Figure 4Db*). The latter effect may result from defects in vesicle replenishment rates, either through the impaired physical movement of vesicles or disrupted site clearance (*Sakaba and Neher, 2003*; *Lee et al., 2012*; *Hallermann and Silver, 2013*; *Miki et al., 2016*), as $Ca^{2+}$-dependent vesicle refilling mediates facilitation and augmentation (*Figure 2* and *Figure 3*). Meanwhile, the former effect might be attributed to a shift in equilibrium in the priming states toward a loosely docked state (LS) at rest. It is unknown whether actin affects vesicular fusogenicity in neurons, although it facilitates multiple steps of vesicle fusion by enhancing membrane tension in neuroendocrine cells (*Wu and Chan, 2022*).

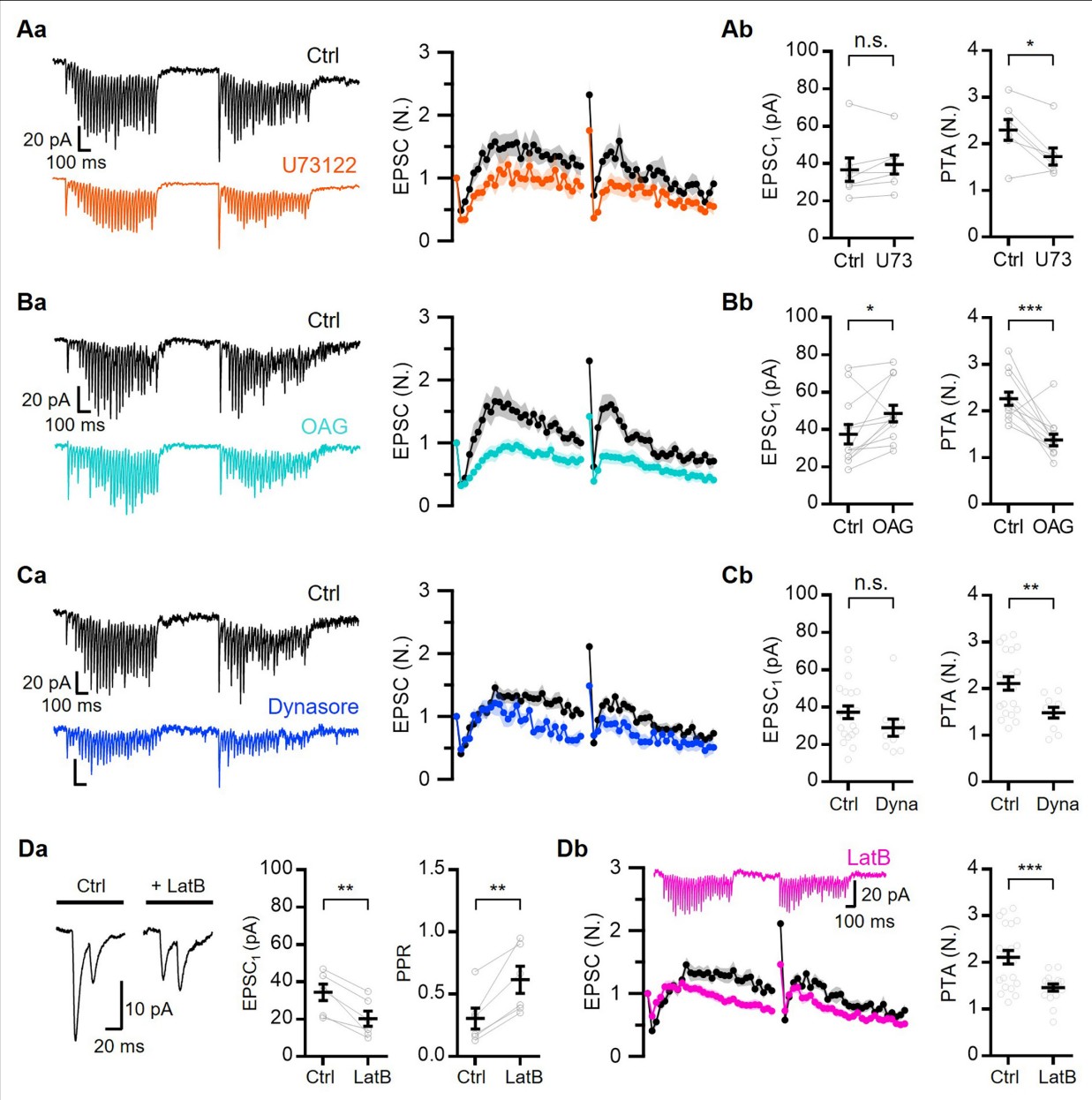

**Figure 4.** Pharmacological experiments reveal specific molecular mechanisms underlying vesicle loading processes. (**A–C**) (**a**) Representative EPSC traces (*left*) and mean baseline-normalized EPSCs (*right*) evoked by double 40 Hz train stimulations separated by 0.5 s inter-burst interval (IBI) in control and in the presence of 5 µM U73122 (Aa, n=7, *orange*), 20 µM OAG (Ba, n=12, *cyan*) or 100 µM dynasore (Ca, n=10, *blue*). (**b**) Mean values for baseline EPSCs (EPSC$_1$, *left*) and augmentation (*right*) from the experiments shown in the corresponding 'a' panel. (Da) Representative EPSC traces evoked by paired pulses (*left*) and mean values for baseline EPSC amplitude (*middle*) and paired pulse ratio (PPR) (*right*) before and after applying 20 µM LatB (n=6). (Db) Representative EPSC traces (*left, upper*) and average of normalized EPSCs (*left, lower*) evoked by double 40 Hz train stimulation separated by 0.5 s in control (n=21) and in 20 µM LatB conditions (n=16; *pink*). *Right*, Mean values for augmentation in control and LatB conditions. *Gray symbols*, individual data. All statistical data are represented as mean ± SEM, *, p<0.05; **, p<0.01; ***, p<0.001; unpaired or paired t-test; n.s.=not significant.

The online version of this article includes the following source data for figure 4:

**Source data 1.** Effects of drugs and Sy7-KD on STP and PTA at L2/3 excitatory synapses.

## STF at both types of local excitatory synapses in L2/3 is abolished by Syt7 KD

We examined whether Syt7 mediates facilitation at the PC-PC and PC-FSIN synapses in L2/3 of the mPFC. To this end, oChIEF and short hairpin RNA (shRNA) against Syt7 (shSyt7) were co-expressed in

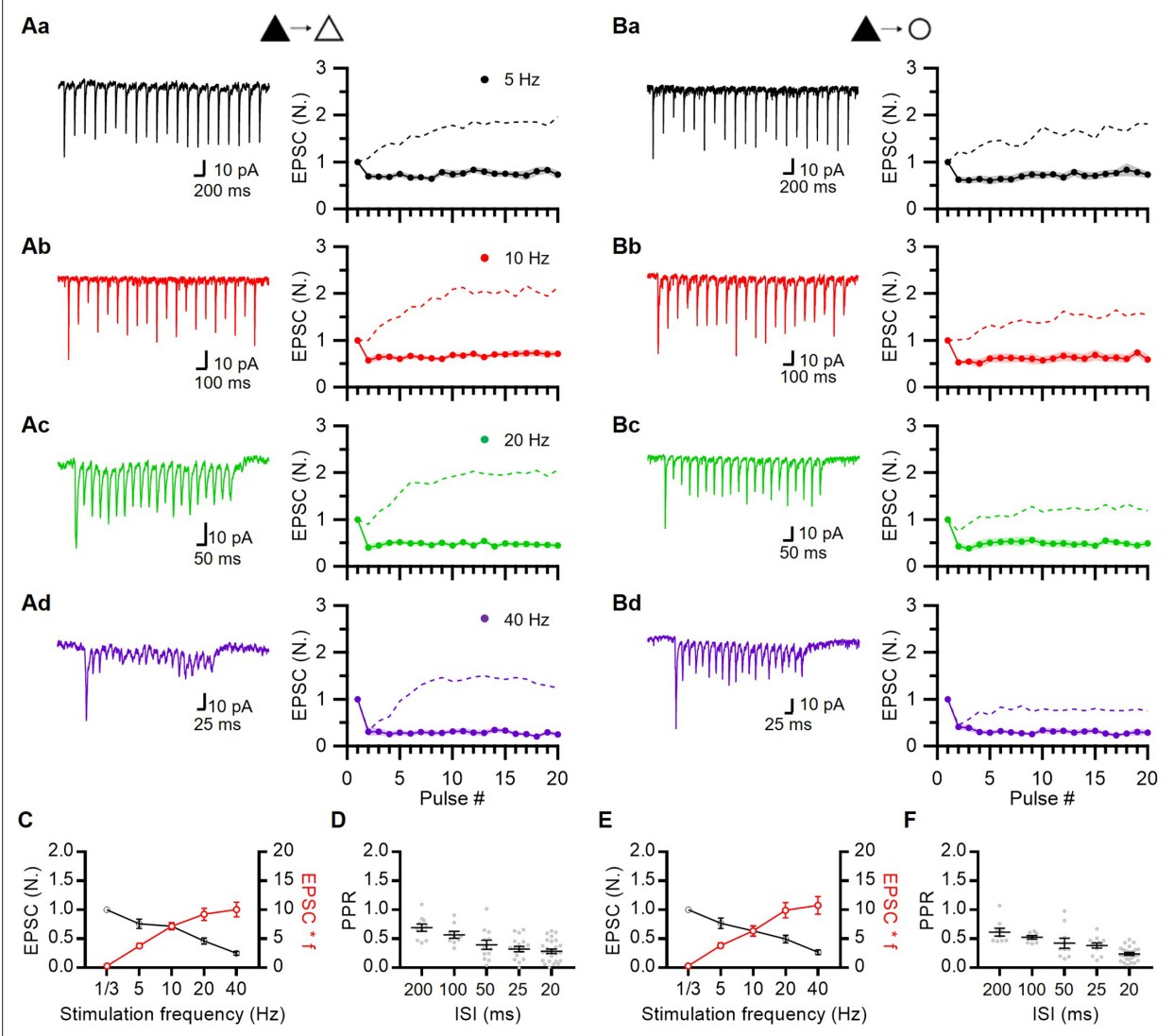

**Figure 5.** Short-term facilitation (STF) at both types of local excitatory synapses is abolished by Syt7 knockdown (KD). (**A, B**) Representative EPSC traces (*left*) and mean baseline-normalized amplitudes of EPSCs (*right*) evoked by 20-pulse trains at 5–40 Hz. Short-term synaptic plasticity (STP) was measured at PC-PC (A; n=10, 9, 12, 9) and PC-FSIN (B; n=9, 8, 10, 9) synapses, in which presynaptic Syt7 transcripts were depleted (Syt7 KD). Syt7 KD pyramidal cells are indicated as *black triangles* on the top. For comparison, STP in wild-type (WT) synapses is reproduced from **Figure 1** (*dotted lines*). Same frequency color codes were used as in **Figure 1**. (**C**) Baseline-normalized amplitudes of steady-state EPSC (EPSC$_{ss}$; *black symbols*) and synaptic efficacy (EPSC$_{ss}$ ×f; *red symbols*) as a function of stimulation frequency (**f**) at PC-PC synapses. EPSC$_{ss}$ was defined as the average of last five EPSC amplitudes from 20-pulse trains. (**D**) Paired pulse ratio (PPR) as a function of inter-spike intervals (n=10, 9, 12, 18, 26). (**E**) EPSC$_{ss}$ and synaptic efficacy at PC-FSIN synapses. (**F**) PPR at PC-FSIN synapses (n=9, 8, 10, 12, 24). *Gray symbols*, individual data.

The online version of this article includes the following source data and figure supplement(s) for figure 5:

**Source data 1.** STP at Syt7-KD synapses.

**Figure supplement 1.** Little effects of Syt7 KD on intrinsic properties.

**Figure supplement 2.** Presynaptic expression of shRNA-insensitive Syt7 rescues facilitation at L2/3 excitatory synapses.

**Figure supplement 2—source data 1.** STP at KD and rescued synapses (**Figure 5—figure supplement 2B & E**).

L2/3 PCs using IUE. Syt7 KD resulted in a complete loss of facilitation at all tested stimulation frequencies (**Figure 5A and B**). Short-term depression and paired-pulse depression were more pronounced as the stimulation frequency increased, and the frequency invariance of steady-state EPSC was abolished, implying little contribution of Ca$^{2+}$-dependent vesicle recruitment (**Figure 5C–F**). The lack of facilitation could not be attributed to the failure of APs in presynaptic axons, as optical stimulation elicited reliable APs in Syt7 KD PCs during the 40 Hz train (**Figure 1—figure supplement**

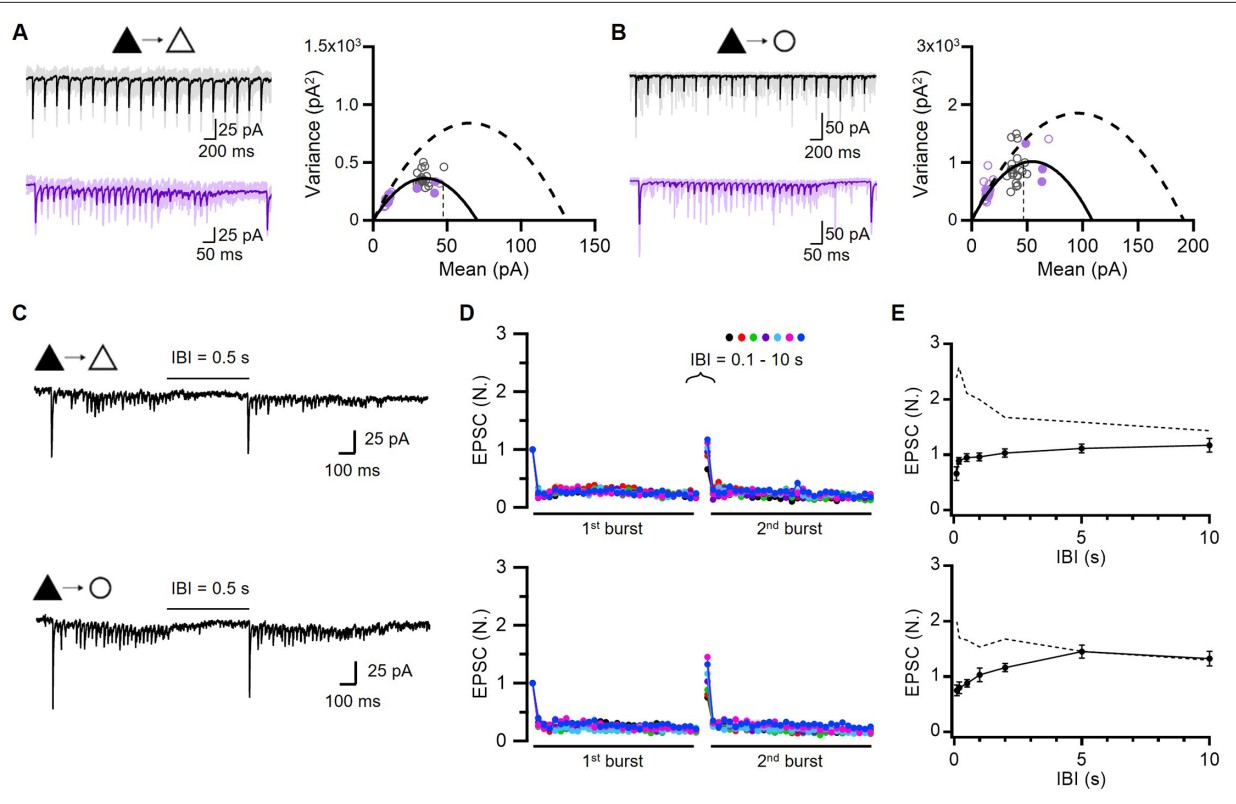

**Figure 6.** Syt7 knockdown (KD) synapses exhibit complementary changes in the number of release sites and their vesicle occupancy. (**A, B**) *Left*, Representative traces of EPSCs evoked by 5 and 40 Hz train stimulations (*black*, 5 Hz; *purple*, 40 Hz). *Right*, Variance-mean plots of EPSCs amplitude from averaged EPSCs recorded at PC-PC (A; n=9, 15, 12, 10, 9) and PC-FSIN (B; n=4, 14, 10, 9, 8) synapses in which presynaptic Syt7 has been knocked down. The data were fitted using multiple-probability fluctuation analysis (MPFA), and error bars are omitted for clarity. The mean 1st EPSC amplitude of 5 Hz train (*vertical broken line*) was used for estimation of baseline $p_{occ}$. Dashed parabolas indicate MPFA fits to variance-mean plot of wild-type (WT) synapses reproduced from *Figure 3*. (**C–E**) Recovery experiments at PC-PC (*top*) and PC-FSIN (*bottom*) synapses in which presynaptic Syt7 was knocked down. (**C**) Representative EPSCs evoked by double 40 Hz train stimulations separated by 0.5 s. (**D**) Mean baseline-normalized amplitudes of EPSCs evoked by double 40 Hz trains at PC-PC (n=12, 10, 9, 9, 10, 9, 8) and PC-FSIN (n=10, 9, 10, 11, 8, 9, 10) synapses at different inter-burst intervals (IBIs). (**E**) Recovery time course. Baseline-normalized amplitudes of 1st EPSC from the 2nd burst were plotted as a function of various IBIs. *Dotted lines* indicate augmented EPSCs in the WT reproduced from *Figure 3*.

The online version of this article includes the following source data for figure 6:

**Source data 1.** Post-tetanic augmentation at Syt7-KD synapses.

**Source data 2.** Variance-mean plots of EPSCs at Syt7-KD synapses.

2C). Syt7 KD had little effect on the basal properties, including the peak amplitude, rise/decay time, and time-to-peak, of single AP-induced EPSCs (*Figure 1—figure supplement 3D and E*). Moreover, Syt7 KD did not significantly affect the intrinsic properties of neurons (*Figure 5—figure supplement 1*).

To examine whether Syt7 KD had any nonautonomous effects on postsynaptic cells, we measured the quantal size (q) from asynchronous events in the presence of $Sr^{2+}$ and found that the mean quantal size remained unaltered at Syt7 KD synapses (*Figure 3—figure supplement 1C*). This indicated that Syt7 did not influence postsynaptic parameters, including the density and kinetic properties of AMPARs. To test the specificity of shSyt7, we examined whether co-expression of the shRNA-resistant form of Syt7 restored synaptic facilitation in both excitatory synapse types. Presynaptic overexpression of shRNA-resistant Syt7 under the CAG promoter rescued the effect of shRNA at both PC-PC and PC-FSIN synapses (*Figure 5—figure supplement 2*). These results underscore the crucial role of presynaptic Syt7 in mediating STF at local excitatory synapses in L2/3 of the mPFC.

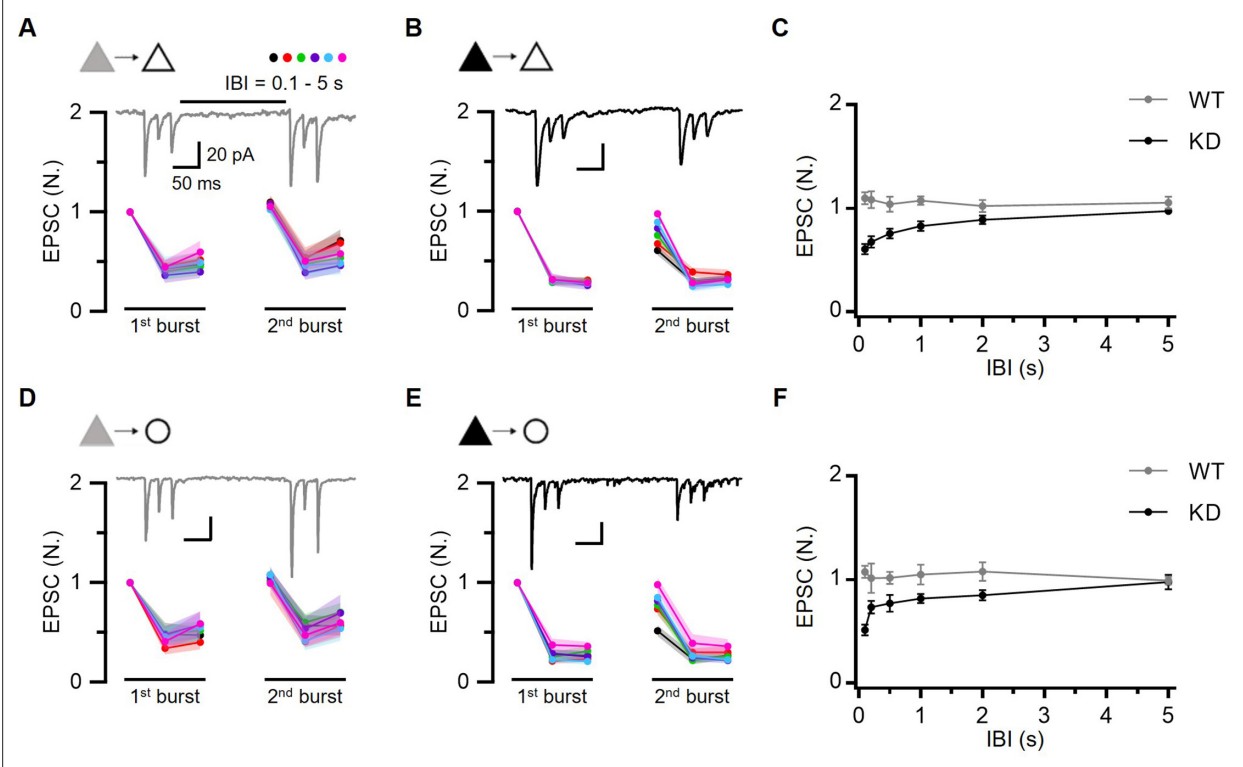

**Figure 7.** Recovery of tightly docked state (TS) vesicles following depletion is accelerated by Syt7. (A–C) Recovery experiments at PC-PC synapses in wild-type (WT) (A) and Syt7 knockdown (KD) (B). Mean baseline-normalized amplitudes of EPSCs evoked by two consecutive 3-pulse 40 Hz trains in WT (n=16, 13, 11, 11, 11, 9 from short to long inter-burst intervals [IBIs], respectively) and KD (n=13, 11, 12, 8, 10, 11) synapses at different IBIs (0.1, 0.2, 0.5, 1, 2, 5 s). *Inset*, representative traces of EPSCs evoked by two consecutive 3-pulse 40 Hz train stimulations separated by 0.2 s. (D–F). Recovery experiments at PC-FSIN synapses in WT (D) and Syt7 KD (E). Mean baseline-normalized amplitudes of EPSCs evoked by two consecutive 3-pulse 40 Hz trains in WT (n=9, 11, 11, 9, 10, 13) and KD (n=9, 12, 9, 7, 10, 9) synapses at different IBIs. *Inset*, representative traces of EPSCs evoked by two consecutive 3-pulse 40 Hz train stimulations separated by 0.2 s. (C, F) Recovery time course. Baseline-normalized amplitudes of 1st EPSC from the 2nd burst were plotted as a function of various IBIs.

The online version of this article includes the following source data for figure 7:

**Source data 1.** Recovery of TS vesicles following depletion is accelerated by Syt7.

## Syt7 KD synapses exhibit complementary changes in the number of release sites and their vesicle occupancy

To estimate the quantal parameters in Syt7 KD synapses, we performed V-M analysis of EPSCs evoked by train stimulation (*Figure 6A and B*). Fitting a parabola to the V-M plot revealed higher baseline $p_{occ}$ in Syt7 KD synapses than in wild-type (WT) synapses for both synapse types (KD, 0.68 and 0.42; WT, 0.32 and 0.31 for PC-PC and PC-FSIN synapses, respectively). The $p_{occ}$ monotonously decreased during a train stimulation, as expected from short-term depression. Moreover, the number of release sites (N) was markedly lower in Syt7 KD synapses (N=3.5) than in WT synapses (N=5.3; *Figure 3A*). Next, we examined the PTA induced by the same protocol as in *Figure 3* (two 40 Hz train stimulations separated by variable IBIs) at Syt7 KD synapses (*Figure 6C*). The 40 Hz train-induced synaptic depression quickly recovered during IBIs, but no PTA was observed at Syt7 KD synapses, indicating its crucial role in augmentation (*Figure 6D and E*).

## TS vesicle recovery following depletion is accelerated by Syt7

Syt7 is recognized not only for its crucial role in facilitation, but also for mediating Ca²⁺-dependent replenishment of releasable vesicles (*Liu et al., 2014*; *Tawfik et al., 2021*). Given that facilitation at L2/3 excitatory synapses is driven by progressive overfilling (*Figure 2* and *Figure 3*), the loss of facilitation at Syt7 KD synapses in the present study implies that Syt7 plays a key role in the Ca²⁺-dependent acceleration of refilling and overfilling of the DS with TS vesicles. To test this hypothesis,

we examined the recovery kinetics of EPSCs after TS vesicles were depleted by 3 pulses at 40 Hz in WT and Syt7 KD synapses (*Figure 7*). Strong PPD was observed not only in the first burst, but also in the second burst. This suggested that EPSCs in the second burst were mediated by TS vesicles, similar to those in the first burst (*Figure 7A and D*). Therefore, $EPSC_1$ in the second burst can be interpreted as a release from TS vesicles recovering during a given IBI. The recovery time course of $EPSC_1$ in the second burst indicated full recovery of TS vesicles within 100 ms in both excitatory synapse types in WT rats (*Figure 7C and F*). However, this recovery was remarkably slower in Syt7 KD synapses, requiring 5 s for full recovery, indicating that TS vesicle recovery was greatly accelerated by Syt7. Overall, synaptic facilitation and augmentation at local excitatory synapses in L2/3 can be ascribed to progressive activity-dependent increases in the occupancy of TS vesicles (*Figure 2* and *Figure 3*), in which Syt7 plays a key role.

## Behavioral consequences of Syt7 KD in L2/3 PCs of the mPFC

We previously proposed that the disparity in STP between the PC-PC and PC-IN synapses results in activity-dependent changes in the ratio of synaptic weights at the PC-PC and PC-IN synapses ($J_{ee}/J_{ie}$). This may, in turn, have a profound influence on persistent activity in a recurrent network by biasing the E-I balance (*Yoon et al., 2020*). Syt7 KD not only eliminated facilitation, but also the activity-dependent increase in the $J_{ee}/J_{ie}$ ratio in the L2/3 network (*Figure 8—figure supplement 1*). This suggests the possibility that Syt7 contributes to the persistent activity during working memory tasks in the L2/3 recurrent network. tFC requires associative learning between an auditory cue (conditioned stimulus, CS) and temporally separate aversive events (unconditioned stimulus, US). The formation of trace fear memory requires the prelimbic area of the mPFC and hippocampus. mPFC activity during the trace interval is thought to temporarily hold CS information, enabling the network to associate temporally discontinuous events (i.e. temporal associative learning) (*Gilmartin et al., 2013*). Synaptic facilitation and augmentation are implicated in temporary memory retention in recurrent networks (*Mongillo et al., 2008*; *Mongillo et al., 2012*).

Given that Syt7 is crucial for synaptic facilitation (*Figure 5*) and augmentation (*Figure 6*) in prelimbic L2/3 recurrent excitatory synapses, we tested whether trace fear memory formation was impaired in rats with depleted Syt7 transcripts, specifically in the L2/3 PCs of the mPFC. Syt7-targeted shRNA (shSyt7) or scrambled shRNA (Scr) constructs were transfected bilaterally into the L2/3 PCs of the mPFC using IUE (*Figure 8A*). Both the Scr-transfected control and shSyt7-transfected rats underwent tFC training (*Figure 8B*). Consistent with *Gilmartin et al., 2013*, in which prefrontal persistent firing was optogenetically inhibited, WT and KD animals exhibited similar freezing behavior during this acquisition phase. The following day, the rats were subjected to a tone test in which trace fear memory formation was assessed from the freezing response to a tone alone in a distinct context. Compared to the control group, the shSyt7-transfected group exhibited significantly lower levels of freezing behavior in response to auditory cues during the first four trials, suggesting an impairment in trace memory formation (*Figure 8C*).

The tone test was repeated the following day to test for extinction memory formation. Syt7 KD animals exhibited much less freezing to the CS (+1d) than did WT animals. Considering the Rescorla-Wagner model (*Yau and McNally, 2023*), our results suggest that the strength of CS-US associative memory was weaker in KD animals than in WT animals; thus, the acquisition of trace fear memory was impaired by Syt7 KD in the L2/3 PCs of the mPFC.

Next, we assessed the locomotor activity and anxiety levels of rats using the open field test (OFT) and elevated plus maze (EPM) test. Scr controls and Syt7 KD rats showed similar exploratory patterns, displaying comparable total distance moved and time spent in the periphery of the OFT or the open/closed arms of the EPM (*Figure 8—figure supplement 2*). Previous studies have suggested that the acquisition of trace fear memory requires elevated trace activity in prelimbic PCs (*Gilmartin et al., 2013*). Given that c-Fos, an immediate early gene, is upregulated following patterned activities in neurons, its expression is considered a reliable indicator of recent neuronal activation (*Fields et al., 1997*; *Guzowski et al., 2005*). To investigate the effect of Syt7 on neuronal activity in vivo during tFC, animals were sacrificed at the time of peak c-Fos protein expression, approximately 90 min after completion of the behavioral task (*Arime and Akiyama, 2017*).

To elucidate the activation patterns of specific neuronal populations, we performed immunohistochemical co-labeling of c-Fos and GAD67 (a GABAergic interneuron marker) upon tFC acquisition.

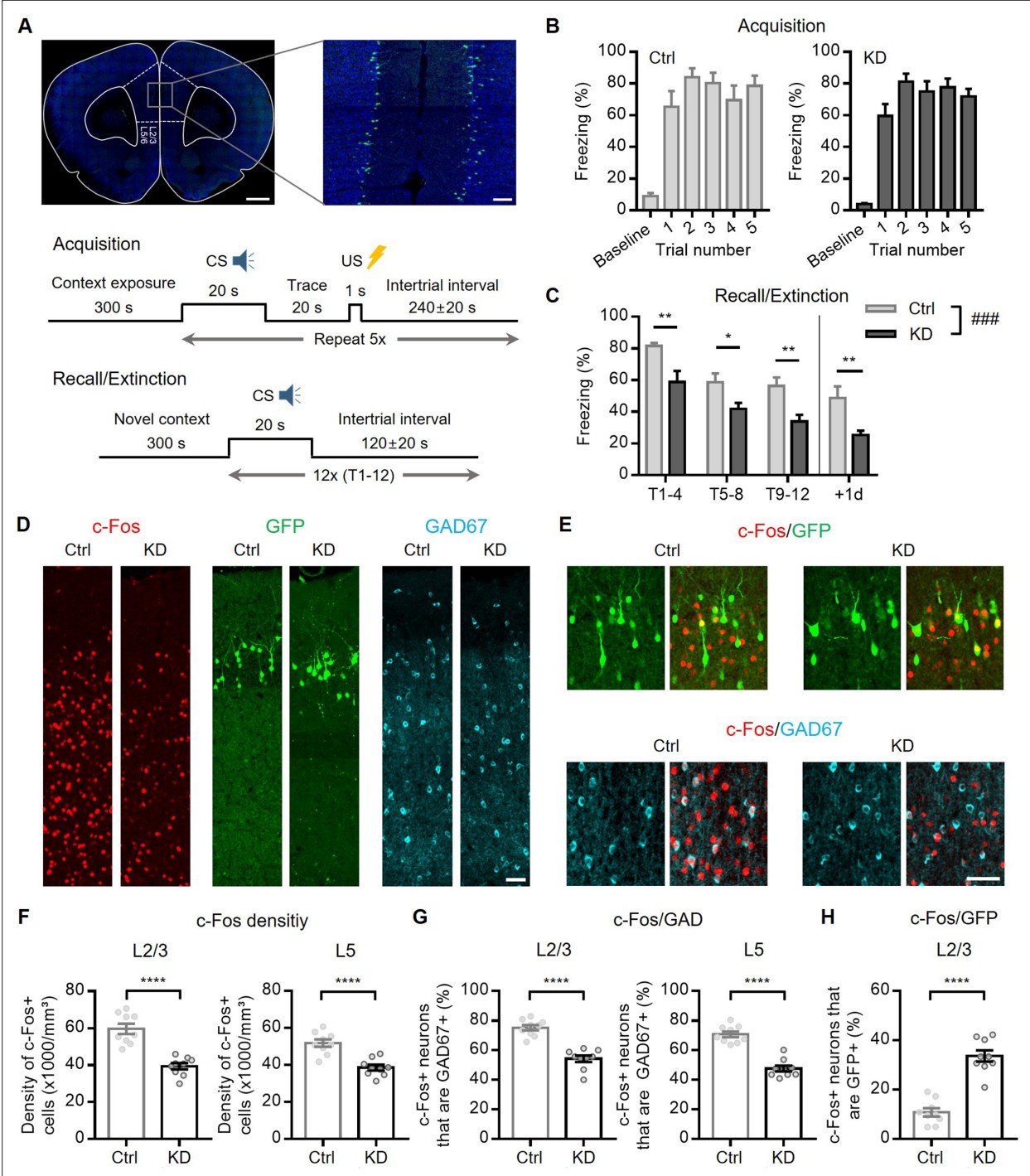

**Figure 8.** Behavioral effects of Syt7 deficiency in layer 2/3 (L2/3) pyramidal cells (PCs) of the medial prefrontal cortex (mPFC). (**A**) *Top,* Representative images showing bilateral expression of U6-GFP in L2/3 of PCs after in utero electroporation (IUE) at embryonic day (E)17.5. Scale bar: 1 mm, 100 μm. *Bottom,* Schematic of trace fear conditioning and extinction (tone test) protocol. (**B**) Freezing behavior of control (expressing scrambled short hairpin RNA [shRNA], *Scr*) and Syt7 knockdown (KD) rats during acquisition of trace fear conditioning (tFC). (**C**) Freezing ratio during tone tests on following days. Data are shown as average freezing during T1–T4, T5–T8, or T9–T12 (T, trials). The freezing on T1-T4 was significantly lower in KD rats, suggesting that formation of trace memory was impaired in Syt7 KD rats (n=10, 11 for Ctrl and KD, respectively; p=0.0007, F(1, 19)=16.43, two-way repeated measures ANOVA; p=0.0064, 0.0211, 0.0064, 0.0064; Holm-Sidak test). (**D**) Representative images of c-Fos immunoreactivity in the prelimbic cortex of control or Syt7 KD rats 90 min after tFC acquisition. c-Fos (*left, red,* Cy5) and GAD67 (*right, cyan,* Cy3) were immunostained in the same brain slice expressing U6-GFP (*middle; green*). Scale bar, 50 μm. (**E**) Exemplar images of c-Fos positive neurons expressing GFP (*top*) or GAD67 (*bottom*) in control (*left*) or Syt7 KD rats (*right*). Scale bar, 50 μm. (**F–H**) Effects of Syt7 KD on c-Fos density (**F**) and percentage of c-Fos positive neurons co-labeled with

*Figure 8 continued*

GAD67 (**G**) or GFP (**H**) in L2/3 or layer 5 (L5) of prelimbic cortex (n=9, 9 for Ctrl and KD, respectively). *Open symbols*, individual data. All statistical data are represented as mean ± SEM; ****, p<0.0001; unpaired t-test.

The online version of this article includes the following source data and figure supplement(s) for figure 8:

**Source data 1.** Trace fear conditioning and retrieval in control and KD animals (*Figure 8B-C*).

**Figure supplement 1.** Activity-dependent increase in the synaptic weight ratio at PC-PC to at PC-FSIN connections (Jee/Jie) is abolished by Syt7 KD.

**Figure supplement 2.** Locomotor activity and anxiety levels in Syt7 KD rats using OFT and EPM.

**Figure supplement 2—source data 1.** Little effects of Syt7 deficiency in L2/3 PCs of the mPFC on locomotor and anxiety behaviors.

The density of c-Fos(+) cells and the percentage of c-Fos(+) cells among GAD67(+) cells in all prelimbic layers were significantly lower in Syt7 KD rats than in control rats (*Figure 8F and G*). Additionally, analysis of the proportion of active cells among transfected L2/3 PCs (c-Fos+/GFP+) revealed that Syt7 KD L2/3 PCs were significantly more active than Scr-transfected PCs (*Figure 8H*). This unexpected hyperactivity of Syt7 KD PCs may be attributed to the high reciprocal connectivity between PCs and FSINs in the neocortex (*Holmgren et al., 2003*; *Otsuka and Kawaguchi, 2009*), as PCs with short-term depression would transmit less input to FSINs, which, in turn, would exert less feedback inhibition on reciprocally connected PCs. Collectively, these findings show that the slow activation of vesicle refilling supports various forms of facilitation at excitatory synapses in PFC L2/3, which is essential for neuronal activity during temporal associative learning.

## Discussion

The present study characterized STP at local excitatory synapses in the L2/3 network of the rat mPFC at physiological extracellular [Ca$^{2+}$]. These synapses displayed initial strong depression and then delayed facilitation at 40 Hz, while monotonous slowly developing facilitation was observed at lower frequencies. Our study suggests that such delayed facilitation after a brief depression results from a high $p_v$ (*Figure 2* and *Figure 3* and *Figure 3—figure supplement 2*), along with Ca$^{2+}$-dependent delayed acceleration of vesicle refilling and overfilling (*Figure 2* and *Figure 3*). The twofold increase in EPSC during 20-pulse train stimulations under the condition of such a high $p_v$ suggests a Ca$^{2+}$-dependent increase in the vesicle replenishment rate during a train; this then leads to overfilling of release sites beyond their incomplete basal occupancy (*Figure 3A–B*). The high $p_v$ and strong PPD did not result from artificial bouton stimulation, AMPAR desensitization, or any potential distortion caused by optogenetic methods (*Figure 1—figure supplements 4 and 5*). Additionally, similar to STF, PTA could be explained by an increase in the TS vesicle occupancy (*Figure 3C–E*), consistent with previous studies showing post-tetanic increases in the $p_{occ}$ and/or vesicle pool size at various types of synapses (*Lee et al., 2010*; *Vandael et al., 2020*; *Tran et al., 2023*; *Silva et al., 2024*). These findings suggest that release sites are partially occupied at rest and that a progressive overfilling of release sites with TS vesicles underlies facilitation and augmentation at intracortical L2/3 excitatory synapses.

### Definition of vesicular release probability in the present and previous studies

Previous studies for PTA (*Stevens and Wesseling, 1999*; *Garcia-Perez and Wesseling, 2008*) observed invariance of the RRP size after tetanic stimulation. In these studies, the RRP size was estimated by hypertonic sucrose solution or as the sum of EPSCs evoked 20 Hz/60 pulses train (denoted as 'RRP$_{hyper}$'). Because reluctant vesicles can be quickly converted to TS vesicles (16/s) and are released during a train (*Lee et al., 2012*), it is likely that the RRP size measured by these methods encompasses both LS and TS vesicles. In contrast, we assert high $p_v$ based on the observation of strong PPD, failure rates upon paired stimulations at ISI of 20 ms (*Figure 2* and *Figure 3—figure supplement 2*). Given that single AP-induced vesicular release occurs from TS vesicles but not from LS vesicles, $p_v$ in the present study indicates the fusion probability of TS vesicles. For the same reasons, $p_{occ}$ denotes the occupancy of release sites by TS vesicles. Note that our study does not provide a direct clue whether release sites are occupied by LS vesicles that are not tapped by a single AP, although an increase in the LS vesicle number may accelerate the recovery of TS vesicles. As suggested in *Neher, 2024*, even if

the number of LS plus TS vesicles is kept constant, an increase in $p_{occ}$ (occupancy by TS vesicles) would be interpreted as an increase in 'vesicular release probability' if it was measured based on $RRP_{hyper}$.

By the way, it should be noted that increases in both $p_v$ and $p_{occ}$ may contribute to PTA. Post-tetanic potentiation (PTP) and phorbol esters had similar effects on baseline EPSCs (*Taschenberger et al., 2016*), and PDBu increased both the TS vesicle pool size (1.9-fold) and fusion probability of TS vesicles (1.3-fold) at the calyx synapses (*Lin et al., 2025*). Therefore, it is possible that PTP may increase both $p_{occ}$ and $p_v$ of TS vesicles, although it has not been yet directly tested. Moreover, acceleration of the latency in the hypertonicity-induced vesicle release raises a possibility for an increase in fusion probability of TS vesicles (*Stevens and Wesseling, 1999*; *Garcia-Perez and Wesseling, 2008*), although it is still controversial that the acceleration of the release latency represents a reduction in the activation energy barrier for vesicle fusion (*Schotten et al., 2015*).

## Progressive overfilling of DS

Conventional models for facilitation assume an increase in $p_v$ after an AP of RRP vesicles, which had been in a low $p_v$ or reluctant state at rest (*Dittman et al., 2000*; *Turecek et al., 2016*). This facilitation model is unlikely to explain our results, because TS vesicles are predominantly released not only at rest but also in the facilitated state during or after train stimulation, as evidenced by the strong PPD (ISI, 25 ms) during a 5 Hz train (*Figure 2D and E*) and at different intervals after a 40 Hz/30-pulse trains (*Figure 3D*). Moreover, the PPD (ISI, 25 ms) remained pronounced during the early phase of facilitation (*Figure 7*), arguing against the possibility of slowly increasing the $p_v$ of reluctant vesicles. These results suggested that facilitation was mediated by an activity-dependent progressive increase in occupancy of TS vesicles (TS occupancy). Facilitation through an increase in TS occupancy has been termed frequency facilitation, differentiating it from paired pulse facilitation that relies on a residual calcium-dependent transient increase in the $p_v$ of releasable vesicles (*Neher, 2024*).

Assuming an increase in TS occupancy, to attain a twofold increase in EPSC (*Figure 1*), the baseline occupancy of the DS by TS vesicles should be low, as estimated from the V-M plot (ca. 30% in *Figure 3*). However, it remains unclear whether the remaining DS that are not occupied by TS vesicles are truly vacant or are occupied by reluctant LS vesicles that are not released by the AP or AP trains. TS vesicles can be supplied by the transformation of LS to TS vesicles, both of which reside at the same DS (LS/TS model; reviewed in *Neher, 2024*). Alternatively, TS vesicles can be supplied from a distinct replacement pool (RS/DS model; reviewed in *Silva et al., 2021*). Further studies, including ultrastructural analyses of AZs, are required to address this issue.

Given the crucial role of Syt7 in facilitation (*Figure 5*), the facilitation mechanisms are closely related to how Syt7 mediates synaptic facilitation. Based on the slow kinetics of Syt7, previous computational modeling studies have proposed that Syt7 may mediate STF through an activity-dependent increase in $p_v$ (*Turecek et al., 2016*; *Jackman and Regehr, 2017*; *Turecek et al., 2017*; *Norman et al., 2023*). The present study, however, found that both facilitation and augmentation were mediated by overfilling of the release sites (*Figure 2* and *Figure 3*). Moreover, Syt7-KD synapses displayed slower recovery of TS vesicles (*Figure 7*). These results suggest that Syt7 is essential for activity-dependent overfilling of DS with TS vesicles. Whereas the baseline EPSC was not altered (*Figure 1—figure supplement 3*), complementary changes in the number of DS and their baseline occupancy were observed in Syt7 KD synapses (*Figure 6*). These results raise a possibility that Syt7 may play a role in providing additional vacant DS that can be overfilled during facilitation. It remains to be elucidated whether the decrease in the number of DS at Syt7 KD synapses is related to the subcellular localization of Syt7 to the plasma membrane of the AZ (*Sugita et al., 2001*; *Vevea et al., 2021*).

## STP model in light of known Ca²⁺ binding kinetics of Syt7

Syt7 acts as an upstream Ca²⁺ sensor that facilitates activity-dependent replenishment (*Liu et al., 2014*; *Bacaj et al., 2015*; *Chen et al., 2017*; *Tawfik et al., 2021*). The present study shows that EPSCs are mediated by TS vesicles both at rest and during activity-dependent facilitation. Releasable vesicles are homogeneous, and Syt7-dependent overfilling is responsible for the activity-dependent enhancement of EPSCs. Thus, we tested whether slowly developing facilitation could be replicated within the framework of a simple refilling model based on the known Ca²⁺ binding kinetics of Syt7 (*Brandt et al., 2012*). To this end, we made the following assumptions: (1) a Ca²⁺-dependent increase in the forward refilling rate ($k_1$) requires a full Ca²⁺-bound form of Syt7, to which multiple Ca²⁺ ions can bind (two

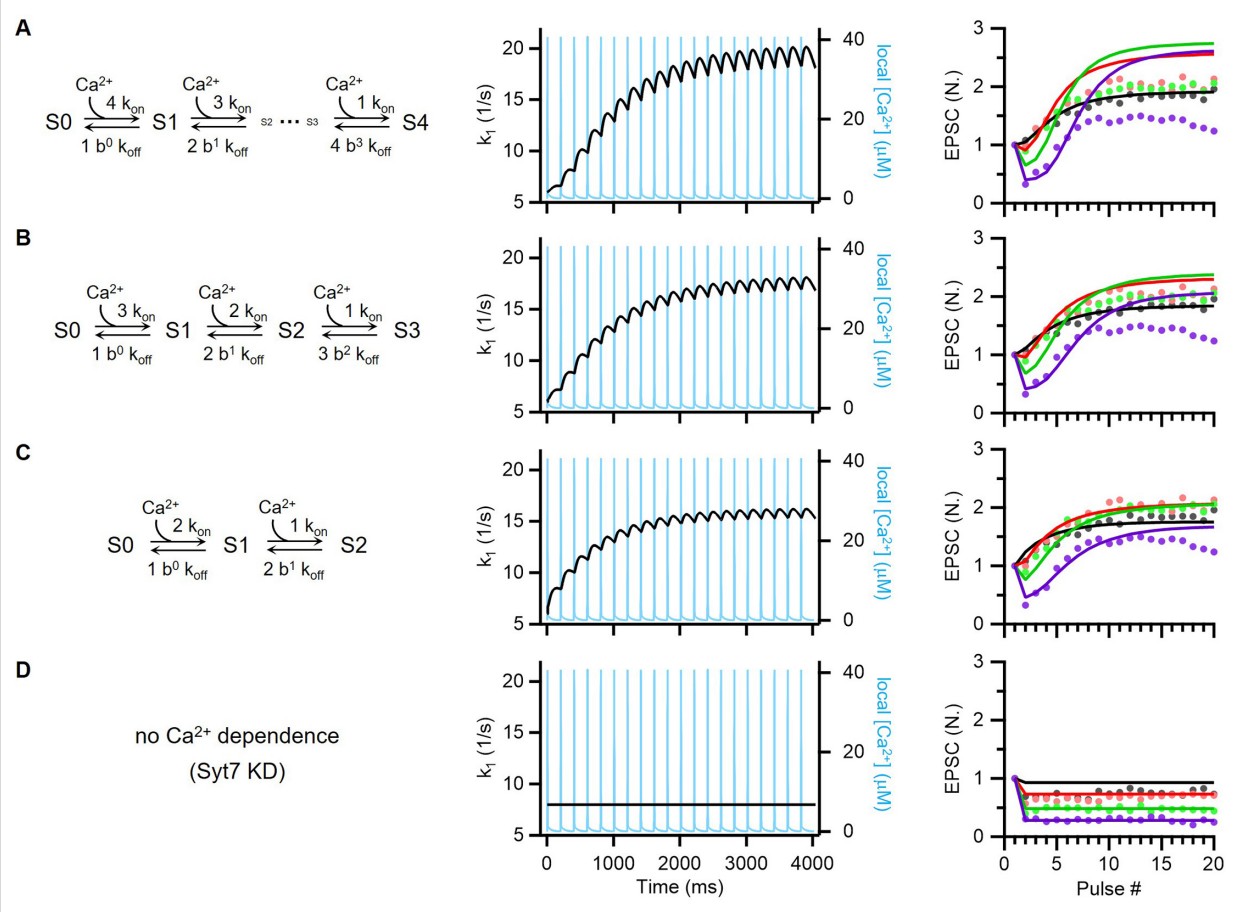

**Figure 9.** Short-term synaptic plasticity (STP) model in light of known $Ca^{2+}$ binding kinetics of Syt7. (**A–D**) *Left*, Schematic of allosteric calcium binding to Syt7. The number of $Ca^{2+}$ bound to Syt7 was denoted as # in 'S#' in the reaction scheme. $k_{on}$ = 7/μM/s, $k_{off}$ = 10/s. *Middle*, Simulated changes of $k_1$ (*black*) in response to 5 Hz train of $Ca^{2+}$ transients (*light blue traces*). The priming step of the simple refilling model was assumed to be catalyzed by the full $Ca^{2+}$-bound form of Syt7. Accordingly, $Ca^{2+}$-dependent increase in $k_1$ was calculated as $K_{1,max}$ multiplied by a fraction of full $Ca^{2+}$-bound form of Syt7. We assumed that local $[Ca^{2+}]_i(t)$ follows a Gaussian function: $(1/\sigma\sqrt{2\pi})\exp[-(t-t_p)^2/2\sigma^2]$, in which $t_p$ = 0.25 ms and σ=0.085 ms. *Right*, Fits of the Syt7 model to the STP data. To fit this model to the STP data, $K_{1,max}$ was set to 300/s (**A**), 220/s (**B**), and 180/s (**C**). Cooperativity factor (**b**) was set to 0.35 (**A**), 0.2 (**B**), and 0.05 (**C**). $k_1$ is set to be constant for Syt7 knockdown (KD) (**D**). Same frequency color codes were used as in *Figure 1*.

The online version of this article includes the following source data for figure 9:

**Source data 1.** Matlab codes for simulation in *Figure 9*.

or three $Ca^{2+}$ ions on each C2A and/or C2B domain) (*Brandt et al., 2012*; *Chon et al., 2015*; *Voleti et al., 2017*); (2) sequential $Ca^{2+}$ binding steps are synergistic through a cooperativity factor, b; (3) Syt7 senses AP-induced local $[Ca^{2+}]$ similar to that estimated at the calyx of Held (a Gaussian function with a full width at half maximum value of 0.2 ms and a peak of 40 μM) (*Wang et al., 2008*); and (4) residual $[Ca^{2+}]$ follows a mono-exponential function with an amplitude of 1 μM and decay time constant of 50 ms (*Jackson and Redman, 2003*).

Under these assumptions and the framework of the simple refilling model, slow and delayed facilitation could be replicated by modeling $k_1$ as $k_1 = k_{1,b} + K_{1,max} \times$ [Syt7: n $Ca^{2+}$], where [Syt7: n $Ca^{2+}$] represents the fraction of fully $Ca^{2+}$-bound Syt7 (*Figure 9A–C*). Syt7 was slowly activated over the course of the stimulus train, suggesting that the cooperative binding of multiple $Ca^{2+}$ ions progressively increased the putative active form of Syt7 because of its slow membrane binding/unbinding kinetics. Nevertheless, the late phase of facilitation at high frequencies was predicted to be higher in the simulation than in the experiment. As expected, based on the notion that sustained release during HFS is limited by site clearance in the AZ (*Figure 4C*), the model/data discrepancy became more pronounced at higher frequencies. Incorporating endocytic terms into the model may mitigate

late-phase discrepancies. Moreover, removal of the catalytic function of Syt7 in the model effectively accounted for the complete loss of facilitation observed in Syt7 KD (*Figure 9D*).

## Conventional STF models involving cumulative increase in $p_v$ of reluctant vesicles

Conventional models for facilitation assume a post-AP residual $Ca^{2+}$-dependent step increase in $p_v$ of RRP (*Dittman et al., 2000*) or reluctant vesicles (*Turecek et al., 2016*). Given that $p_v$ of TS vesicles is close to one in the present study, an increase in $p_v$ of TS vesicles cannot account for facilitation. The possibility for activity-dependent increase in fusion probability of LS vesicles (denoted as $p_{v,LS}$) should be considered in two ways depending on whether LS and TS vesicles reside in distinct pools or in the same pool. Notably, strong PPD at short ISI implies that $p_{v,LS}$ is near zero at the resting state. Whereas LS vesicles do not contribute to baseline transmission, STF may be mediated by cumulative increase in $p_v$ of LS vesicles that reside in a distinct pool. Because the increase in $p_{v,LS}$ during facilitation recruits new release sites (increase in N), the variance of EPSCs should become larger as stimulation frequency increases, resulting in upward deviation from a parabola in the V-M plane, as shown in recent studies (*Valera et al., 2012*; *Kobbersmed et al., 2020*). This prediction is not compatible with our results of V-M analysis (*Figure 3*), showing that EPSCs during STF fell on the same parabola regardless of stimulation frequencies. Therefore, it is unlikely that an increase in fusion probability of reluctant vesicles residing in a distinct release pool mediates STF in the present study. For the latter case, in which LS and TS vesicles occupy the same release sites, it is hard to distinguish a step increase in fusion probability of LS vesicles from a conversion of LS vesicles to TS. Nevertheless, our results do not support the possibility for a gradual increase in $p_{v,LS}$ that occurs in parallel with STF. Strong PPD, indicative of high $p_v$, was consistently found not only in the baseline (*Figure 2* and *Figure 2—figure supplement 1*) but also during the PTA phase (*Figure 3D*) and even during the early development of facilitation (*Figure 2D and E* and *Figure 7*), arguing against the gradual increase in $p_{v,LS}$. One may argue that STF may be mediated by a drastic step increase of $p_{v,LS}$ from zero to one, but it is not distinguishable from conversion of LS to TS vesicles.

## Possible mechanisms underlying the depression-facilitation sequence

Cooperative binding of multiple $Ca^{2+}$ ions to Syt7 predicts delayed activation and a progressive increase in vesicle refilling during train stimulations, resulting in delayed facilitation. Combining this with a high $p_v$ could replicate not only the slow facilitation at 5 Hz, but also the unique depression-facilitation sequence observed at 40 Hz (*Figure 9*). As simulated by *Pulido and Marty, 2018*, delayed facilitation can be reproduced by a high $p_v$ in combination with a low baseline occupancy of RS and subsequent $Ca^{2+}$-dependent refilling. However, whether this model is also able to replicate slow facilitation at low frequencies remains to be tested. An alternative, but not exclusive, possibility for delayed facilitation is that the presynaptic global $Ca^{2+}$ itself may undergo a slow buildup during a train due to the saturation of endogenous $Ca^{2+}$ binding proteins.

## Facilitation requires not only Syt7, but also activation of PLC-DAG signaling

To gain insight into the specific molecular mechanisms underlying the vesicle-loading processes, we examined the effects of pharmacological inhibitors targeting PLC and DAG (*Figure 4*). Both a PLC inhibitor and a DAG analog reduced facilitation and augmentation. This suggests that the progressive overfilling of DS with TS vesicles is supported by the PLC-DAG signaling pathway and probably downstream Munc13 (*Rhee et al., 2002*; *Lou et al., 2008*). Phorbol esters have been recently shown to increase the TS fraction of docked vesicles by enhancing vesicle priming without affecting $Ca^{2+}$ transients or $p_v$ (*Taschenberger et al., 2016*; *Aldahabi et al., 2022*; *Papantoniou et al., 2023*; *Aldahabi et al., 2024*). In line with these findings, signaling involving PLC and DAG may support $Ca^{2+}$-dependent overfilling of TS vesicles by enhancing molecular priming processes.

## Roles of facilitation and augmentation in working memory and related tasks

Previous theoretical studies have proposed that working memory in a recurrent network can be maintained in the form of augmentation at synapses (*Mongillo et al., 2008*) and/or persistent activity of

neurons in a memory ensemble (*Mongillo et al., 2012*). The latter can be supported by STF of recurrent excitatory synapses or disparity in STP between excitatory and inhibitory synapses (*Mongillo et al., 2012*; *Yoon et al., 2020*). However, whether these short-term enhancements in synaptic efficacy contribute to working memory-related behaviors has not been examined. tFC is a temporal associative learning task requiring temporary retention of CS information during a trace period. The mPFC and hippocampus are crucial for tFC, and tFC is interfered with by a load of working memory and attention distraction, implying that the tFC shares the same neural substrate with other working memory tasks (*Raybuck and Lattal, 2014*). In the present study, tFC in WT and Syt7 KD animals revealed that Syt7 deficiency in the L2/3 PCs of the mPFC impaired the acquisition of trace fear memory and reduced c-Fos expression associated with tFC training (*Figure 8*).

The acquisition of trace fear memory was impaired by inhibition of persistent activity in mPFC during the trace period (*Gilmartin et al., 2013*). The similar deficit observed in Syt7 KD animals is consistent with the hypothesis that STF provides bistable ensemble activity in a recurrent network (*Mongillo et al., 2012*). Nevertheless, alternative mechanisms may be responsible for the behavioral deficit. Not only does the recurrent network, but also the long-range loop between the mPFC and the mediodorsal (MD) thalamus plays a critical role in maintaining persistent activity within the mPFC, especially for a delay period longer than 10 s (*Bolkan et al., 2017*). Prefrontal L2/3 is heavily innervated by MD thalamus, and L2/3-PCs subsequently relay signals to L5 cortico-thalamic (CT) neurons (*Collins et al., 2018*). Given that L2/3 is an essential component of the PFC-thalamic loop, loss of STF at recurrent synapses between L2/3 PCs may lead to insufficient L2/3 inputs to L5 CT neurons and failure in the reverberant PFC-MD thalamic feedback loop. Therefore, not only L2/3 recurrent network but also its output to downstream network should be considered as a possible network mechanism underlying behavioral deficit caused by Syt7 KD L2/3.

In summary, activity-dependent synaptic facilitation and augmentation at prelimbic L2/3 recurrent excitatory synapses can be attributed to overfilling of release sites with TS vesicles. Given that the lack of STF in L2/3 PCs impairs trace fear memory acquisition, temporary maintenance of CS information during the trace period may be supported by an increase in the TS occupancy in presynaptic terminals of CS-representing neuronal ensembles.

## Materials and methods
### Animals/subjects
All experiments were carried out on Sprague-Dawley rats of either sex at p28–43. Rats were maintained in temperature-controlled rooms on a constant 12 hr light/12 hr dark cycle and were given access to water and food ad libitum. All animal procedures were approved and performed in accordance with the Institutional Animal Care and Use Committee at Seoul National University (SNU200522-1).

### In utero electroporation
Pregnant Sprague-Dawley rats on embryonic day (E) 17.5 were deeply anesthetized with 5% (vol/vol) isoflurane for the duration of surgery. The uterine horns were exposed by laparotomy and wet with warm sterile phosphate-buffered saline (PBS). The plasmids (0.5–2 µg/µl) together with the 0.01% fast green dye were injected into either the left or the right lateral ventricle or bilateral ventricles of embryos through a thin glass capillary (Narishige PC-10). The embryo's head was carefully held between the forceps-shaped electrodes of 10 mm diameter (CUY65010; NEPAGENE, Japan). Electroporation pulses (50 V for 50 ms) were delivered five times at intervals of 150 ms using a square-wave generator (ECM830; BEX, USA). The uterine horns were placed back into the abdominal cavity, and abdominal muscles and skin were sutured. After the surgery, the animals were recovered under the infrared lamp. For KD experiments in slice electrophysiology, the shRNA constructs were co-electroporated with the pCAG-oChIEF-tdTomato, because of the consistent co-expression of the two constructs in a significant proportion of the transfected cells in vivo (*Bony et al., 2013*). Pups with robust fluorescence signals in the prefrontal cortex of both hemispheres were used for behavior experiments.

## Preparation of vector constructs

For an optic stimulation, pCAG-oChIEF-tdTomato was transfected by IUE. shRNA against Syt7 sequence (GATCTACCTGTCCTGGAAGAG) was expressed under the control of an U6 promoter of pAAV-U6-GFP vector (Cell Biolabs, #VPK-413). This shRNA was shown to effectively knock down Syt7 in cell cultures and in vivo by previous reports from more than one research group (*Bacaj et al., 2013*; *Li et al., 2017*). Scrambled control sequence (TCGCATAGCGTATGCCGTT) was designed by shuffling the recognition region sequences and a BLAST analysis confirmed that the control shRNA sequence had no target gene. For rescue experiments, the cDNA for rat Syt7 (NM_021659) rendered resistant to the shRNA was inserted in the KD vector and their expression was driven by the CAG promoter; the vector also contained a self-cleaving P2A peptide sequence followed by EGFP, which ensures visualization of transfected cells. All the constructs above were verified by DNA sequencing. Plasmid DNA was purified and concentrated under endotoxin-free conditions (QIAGEN EndoFree Maxi Kit).

## Slice preparation

Acute brain slices were prepared by using a VT1200S Vibratome (Leica Microsystems) after isoflurane anesthesia and decapitation. Coronal sections of the mPFC (300 µm in thickness) were cut in ice-cold artificial cerebrospinal fluid (aCSF) composed of (in mM) 125 NaCl, 3.2 KCl, 25 NaHCO$_3$, 1.25 NaH$_2$PO$_4$, 20 D-glucose, 2 NaPyr, 1 MgCl$_2$, and 1.3 CaCl$_2$ (*Yoon et al., 2020*), bubbled with 95% O$_2$ and CO$_2$ (pH = 7.3; 310 mOsm). Slices were allowed to recover for a minimum of 30 min at 36°C and subsequently maintained at room temperature until recording. The composition of aCSF was identical throughout the preparation, incubation, and recording periods unless otherwise stated. For the recordings, the mPFC slices were continuously perfused with oxygenated aCSF at a flow rate of 1.5 ml/min, and the temperature was maintained at 31–32°C using an in-line temperature controller (Warner Instruments).

## Electrophysiology

Whole-cell patch-clamp recordings were performed from PCs and FSINs in L2/3 of the prelimbic mPFC. Neuronal subtypes were initially selected based on morphology under visual guidance by using an upright microscope equipped with infrared differential interference contrast optics (BX51WI, Olympus), and were further identified by their electrophysiological properties (see *Figure 1—figure supplement 1*; *van Aerde and Feldmeyer, 2015*; *Tremblay et al., 2016*). To analyze these intrinsic properties in current-clamp experiments, we measured the following parameters: (1) resting membrane potential (RMP), (2) input resistance (R$_{in}$), (3) sag ratio, and (4) the F-I curve, which plots firing frequencies (F) against the amplitude of injected currents (I). Both cell types exhibited a low sag ratio below 0.1, consistent with findings from other studies (*van Aerde and Feldmeyer, 2015*; *Yoon et al., 2020*). PCs and FSINs were easily distinguished by their unique firing frequency, Rin, spike adaptation, RMP, and morphological characteristics, including the shape of soma and the thickness of apical dendrites. For experiments involving oChIEF-expressing neurons in WT or Syt7 KD rats, whole-cell recordings were performed from the fluorescently labeled pyramidal neurons in L2/3 of the prelimbic cortex using suitable filter sets. Patch pipettes (tip resistance of 2–3 MΩ) were pulled from borosilicate glass and filled with (in mM) 130 K-gluconate, 8 KCl, 10 HEPES, 0.2 EGTA, 20 Na$_2$Pcr, 0.3 Na$_2$GTP, 4 MgATP. Intracellular solutions were adjusted to pH 7.25 and 300–310 mOsm. Neurons were voltage clamped at −78 mV and series resistance (Rs) was continuously monitored during the EPSCs recordings. Experiments were discarded before analysis if the Rs changed by 20% or larger deviation of baseline value. Measurements were not corrected for Rs compensation, bridge balance, or liquid junction potential. Electrophysiological data were low-pass-filtered at 1 kHz (Bessel) and acquired at 20 kHz using a Multiclamp 700B amplifier paired with Digidata 1440A digitizer and Clampex 10.2 software (Molecular Devices).

## Synaptic stimulation

IUE resulted in sparse expression of channelrhodopsin (10–20%) (*Figure 1A*). Exploiting the sparse expression, we employed a minimal optical stimulation method to activate a single or very few excitatory synapses in L2/3. The optic minimal stimulation was performed by our published protocol with minor modifications (*Yoon et al., 2020*). This approach aimed to minimize the risk of synaptic contamination, which could arise from engaging extra terminals that were initially inactive (depolarized below

threshold) but became active by temporal summation during high-frequency trains. It also helped prevent the buildup of depolarization at the synaptic terminals, which might influence the probability of release, as noted by *Jackman et al., 2014*. To achieve this, we employed a collimated digital micro-mirror device-coupled LED (Polygon400; Mightex Systems) to confine 470 nm blue light to a small area with a radius of approximately 3–8 µm (typically 3–4 µm), as measured at the focal plane of a 60× water immersion objective (numerical aperture [NA]=1.0; LUMPlanFL, Olympus) (*Figure 1—figure supplement 3A*). To find the minimal stimulation area, the radius of illumination area was increased from 2 µm by 50% at each step (*Figure 1—figure supplement 3A*) and selected the smallest illumination area that elicited EPSCs. Setting the duration of illumination between 3 and 5 ms (typically 5 ms), photostimulation onto oChIEF-expressing cells reliably induced APs during 600-pulse trains (see *Figure 1—figure supplement 2*).

STP at various stimulus frequencies (5, 10, 20, and 40 Hz) was examined using 20-pulse trains. Trials of train stimulation were repeated with a prolonged interval of 30 s to avoid change in baseline properties. We observed no systematic changes in the amplitudes of the first EPSC of each train, nor did we find significant differences in the PPR across stimulation trials in the same cell. For each stimulus frequency, approximately 10–20 consecutive traces were collected and then averaged across trials to yield a mean EPSC trace. For recovery experiments, two consecutive trains of 3 or 30 stimuli at 40 Hz, with varying time intervals (0.1–10 s), were applied every 10 or 60 s, respectively, at both excitatory synapses. For 600-stimulus trains at 10 Hz, the trial was delivered only once for each cell to avoid possible after-effects caused by prolonged synaptic stimulation (*Hirsch and Crepel, 1990*). Prior to the stimulus trains, the EPSCs were normalized to the mean amplitude of the baseline. For the baseline measurements, paired pulses with various ISIs (20–200 ms) were delivered every 3 s (1/3 Hz) at least 20 times.

For all recordings, picrotoxin (PTX, 10 µM) was included in the recording solution to isolate glutamatergic responses. mPFC PC terminals were entirely glutamatergic as the application of 6-cyano-7-nitroquinoxaline-2,3-dione (CNQX, 20 µM), an AMPAR antagonist, completely abolished the monosynaptic events evoked by photostimulation. A low concentration of PTX was employed to minimize the relieving effects on spike suppression caused by blockade of GABAa receptors. We confirmed that no synaptic response was elicited by electrical stimulation in the presence of both 10 µM PTX and 20 µM CNQX.

In *Figure 1—figure supplements 4A*, 1 µM TTX (1 mM stock solution) and 0.1 mM 4-AP (500 mM stock solution) were added to the bath solution to block APs and restore presynaptic glutamate release, respectively (*Figure 1—figure supplement 4*). In *Figure 1—figure supplement 5A*, 50 µM of cyclothiazide, a positive allosteric modulator of the AMPARs, was included to minimize rapid desensitization of the postsynaptic receptors. In *Figure 1—figure supplements 5B*, 0.5 mM kynurenate was added to bath solution to prevent possible saturation of AMPARs, especially during maximal synaptic facilitation. For pharmacological manipulations of synaptic vesicle dynamics (*Figure 4*), drugs related to vesicle supply, including U73122, OAG, LatB, EGTA-AM, and dynasore at indicated concentrations, were added to the recording solution during baseline measurements or PTA/recovery experiments (IBI = 0.5 s). Chemicals were from Sigma-Aldrich, Merck, or Tocris.

## Measurements of quantal size

Experiments for measuring asynchronous release were performed as previously described (*Yoon et al., 2020*). In brief, we replaced extracellular $Ca^{2+}$ entirely with 2 mM $Sr^{2+}$ to induce asynchronous release. A single stimulus was delivered at 3 s intervals over 100 repetitions, and asynchronous release events were detected during a 100 ms window starting 25 ms after stimulus onset for each trial. The same procedure was also executed under sham conditions (0% light intensity) to serve as a control, thus mitigating contamination from nonspecific spontaneous events and potential errors associated with the detection algorithm. q was determined as the mean value from the first Gaussian function after fitting the EPSC distribution with either a single or double Gaussian function (*Figure 3—figure supplement 1*).

## Estimation of $Ca^{2+}$ kinetics at axonal boutons

Cytosolic $[Ca^{2+}]$ at single axonal boutons was estimated by employing the two-dye, dual-excitation wavelength technique as described by *Oheim et al., 1998*. This method involves measuring the

fluorescence ratio of a Ca$^{2+}$-sensitive dye to that of a Ca$^{2+}$-insensitive dye to correct for dye concentrations. We used Fluo-5F at concentrations of 150, 250, or 500 µM (dissociation constant, $K_d$ = 2.3 µM) as the calcium indicator dye and Alexa Fluor 555 (50 µM) as the reference dye. The fluorophores were from Invitrogen. Both dyes were co-loaded into presynaptic terminals of L2/3 PCs in mPFC through whole-cell patch clamp. Before data acquisition, the dyes were allowed to diffuse to terminal boutons for at least 30 min after the patch break-in, a duration sufficient for the equilibrium of dye concentrations between the soma and its axonal terminals.

Calcium imaging was performed by sequentially exciting Fluo-5F and AF555 at wavelengths of 473 and 559 nm, respectively, using a confocal laser-scanning microscope (FV1200; Olympus, Japan) equipped with a 60× water immersion objective (NA, 0.9; LUMPlanFl; Olympus, Tokyo, Japan). Once a single axonal bouton was identified, lines for scanning were drawn across boutons, oriented perpendicular to the axon (*Figure 2—figure supplement 2A and B*). After baseline fluorescence was measured, line scanning was proceeded while a single or train APs were induced by applying current pulses. Upon stimulation, green fluorescence but not red one increased along the vertical array of lines across the bouton (*Figure 2—figure supplement 2B and C*). Measurements were obtained under current-clamp conditions, with the resting potential regularly monitored. The scanning speed was adjusted to 600–700 Hz by controlling the line length. Line scan for measuring fluorescent transients in a bouton was repeated at least three times and then averaged to enhance the signal-to-noise ratio. To mitigate photobleaching, minimal intensity of the laser power and maximal pinhole size were used. Emitted light was filtered through 490/590 or 575/675 bandpass filters before detection by photomultiplier tubes for green or red fluorescence, respectively.

Fluorescence measurements were converted to [Ca$^{2+}$]$_i$ using the ratio of background-subtracted fluorescence (F = F$_0$ – F$_b$) from two dyes. The fluorescence ratio (R=F$_{green}$/F$_{red}$) in data traces was converted to [Ca$^{2+}$]$_i$ using the following equation:

$$\left[\text{Ca}^{2+}\right]_i = K_d \times \left(R - R_{min}\right) / \left(R_{max} - R\right). \tag{1}$$

Calibration parameters were obtained using an in-cell calibration protocol (*Helmchen, 2011*), wherein the minimum (R$_{min}$) and maximum ratio (R$_{max}$) values were established through the use of intracellular solutions containing 10 mM EGTA or 10 mM [Ca$^{2+}$], respectively.

When Ca$^{2+}$ is extruded with a rate constant, $\gamma$, after an AP-induced short pulse of Ca$^{2+}$ influx to a single compartment, [Ca$^{2+}$]$_i$ initially rises to A$_{Ca}$ and then undergoes a mono-exponential decay with a time constant ($\tau_{ca}$). Under this framework, A$_{Ca}$ and $\tau_{ca}$ depend on Ca$^{2+}$ binding ratios of endo- and exogenous Ca$^{2+}$ buffers (denoted as $\kappa_S$ and $\kappa_B$, respectively) as following equations (*Neher, 1995*; *Kim et al., 2003*):

$$1/A_{Ca} = \left(1 + \kappa_S + \kappa_B\right) / \Delta\left[\text{Ca}^{2+}\right]_T. \tag{2}$$

$$\tau_{Ca} = \left(1 + \kappa_S + \kappa_B\right) / \gamma \tag{3}$$

where $\Delta$[Ca$^{2+}$]$_T$ is an increment in total [Ca$^{2+}$] (free plus bound) elicited by an AP. Ca$^{2+}$ binding ratio indicates the ratio of concentration changes in the Ca$^{2+}$-buffer complex relative to free [Ca$^{2+}$] change, and linearized $\kappa_B$ is calculated from the dye concentration and the change from [Ca$^{2+}$]$_1$ to [Ca$^{2+}$]$_2$ as follows:

$$\kappa_B = \Delta\left[\text{CaB}\right] / \Delta\left[\text{Ca}^{2+}\right] = \left[\text{B}\right]_t K_d / \left[\left(\left[\text{Ca}^{2+}\right]_1 + K_d\right)\left(\left[\text{Ca}^{2+}\right]_2 + K_d\right)\right]. \tag{4}$$

The total dye concentration, [B]$_t$, was taken as [Fluo-5F] in the patch pipette. With $\kappa_B$ determined from *Equation 4* and $\tau_{ca}$ from the decay of a Ca$^{2+}$ transient, $\kappa_S$ can be estimated by plotting $\tau$ vs. $\kappa_B$ and applying a linear fit to the plot, which gives the x-intercept as –(1 + $\kappa_S$). Once determined, $\kappa_S$ can be used to calculate $\Delta$[Ca$^{2+}$]$_T$.

## Optimization of k$_1$ and p$_v$ from failure rates

Let n$_0$=the number of docked SVs before arrival of a first AP; n$_1$=the number of docked SVs remaining after a first AP (i.e. n$_0$ – n$_1$ vesicles are released); n$_2$=the number of docked SVs just before arrival of a 2nd AP (ISI = 20 ms; *Figure 3—figure supplement 2*). If the number of release sites (N), forward and reverse rates for vesicle docking (k$_1$ and b$_1$, respectively), mean release site occupancy (p$_{occ}$), and

vesicular fusion probability upon an AP ($p_v$) are given, we can predict P($n_0$, $n_1$, $n_2$), the probability for [$n_0$, $n_1$, $n_2$], as following steps.

1. Probability for a synapse harboring $n_0$ docked SVs just before arrival of a first AP is calculated as:

$$P(n_0) = C(N, n_0) \times p_{occ}^{n_0} \times (1 - p_{occ})^{(N-n_0)}.$$

2. Probability that $n_1$ vesicles remain after a first AP when the initial number of docked SVs is $n_0$ [P($n_1$|$n_0$)] is calculated as:

$$P(n_1|n_0) = C(n_0, n_1) \times (1 - p_v)^{n_1} \times p_v^{(n_0-n_1)}, if\, n_0 \geq n_1,$$
$$P(n_1|n_0) = 0,\ if n_0 < n_1.$$

3. When forward and backward rate constants are $k_1$ and $b_1$, respectively, probability that $n_2$ vesicles are docked just before arrival of a 2nd AP when $n_1$ SVs remain after a 1st AP (Δt=20 ms) [P($n_2$ | $n_1$)] is predicted as follows: Assuming that docking and undocking follow a Poisson process,

$$P(n_2|n_1) = \sum_{j=0}^{n_1} C(n_1, j) C(o_1, i+j) B^j (1-B)^{n_1-j} F^{i+j} (1-F)^{o_1-i-j} \cdot 1_{0 \leq (i+j) \leq o_1}$$

, where B=1 – exp(–$b_1$ Δt); F=1 – exp(–$k_1$ Δt); i = $n_2$ – $n_1$; $o_1$=N – $n_1$.

4. Combining probabilities calculated from the above three steps, we can calculate P($n_0$, $n_1$, $n_2$):

$$P(n_0, n_1, n_2) = P(n_0) \times P(n_1|n_0) \times P(n_2|n_1)$$

5. Once P($n_0$, $n_1$, $n_2$) is given, we can predict probability for double failure ($P_{00}$) and probability for the 1st success and the 2nd failure ($P_{10}$) as:

$$P_{00} = \sum_{n_0=0}^{N} \sum_{n_1=n_0}^{N} \sum_{n_2=0}^{N} p(n_0, n_1, n_2)(1 - p_v)^{n_2}.$$
$$P_{10} = \sum_{n_0=0}^{N} \sum_{n_1 \neq n_0}^{N} \sum_{n_2=0}^{N} p(n_0, n_1, n_2)(1 - p_v)^{n_2}.$$

6. Probability for the 1st failure and the 2nd success ($P_{01}$) and probability for double success ($P_{11}$) were calculated as:

$$P_{01} = P(F_1) - P(F_1,\ F_2)$$
$$P_{11} = P(S_1) - P(S_1, F_2)$$

, in which P($F_1$) and P($S_1$) are probability for the 1st failure and the 1st success, respectively, and these are calculated as:

$$P(F_1) = P(n_0)\, P(F_1|n_0) = \sum_{n_0=0}^{N} P(n_0)(1 - p_v)^{n_0}$$
$$P(S_1) = 1 - P(F_1).$$

To optimize $k_1$ and $p_v$, we used the MATLAB routine, *fmincon*, in which the sum of squared errors (=observed value minus predicted value) was set as a cost function to be minimized. Lower and upper bounds for $k_1$ and $p_v$ were set as [4, 8] and [0.8, 1], respectively. Release probability ($p_r$) was calculated from the observed 1st failure rate, P($F_1$), as

$$p_r = 1 - P(F_1)^{(1/N)}$$

N was assumed to be six as shown in *Figure 3*. Other parameters ($b_1$ and $p_{occ}$) were set according to the following relationships:

$$p_{occ} = p_r/p_v.$$

$$b_1 = k_1(1 - p_{occ})/p_{occ}.$$

## Calculation of probability for double failure

Double failure probability, $P_{00}$, can be more easily predicted. By the Bayes rule,

$$P(F_1, F_2) = \sum_{n_2=0}^{N} \sum_{n_1=0}^{N} P(F_2|n_2)\, P(n_2|n_0)\, P(F_1|n_0)\, P(n_0)$$

where $F_1$ and $F_2$ mean 1st and 2nd failure, respectively, because $n_0 = n_1$ after the 1st failure. $P(F_1)$ and N are observed values. For calculation in *Figure 3—figure supplement 2*, $k_1$ and $p_v$ were set free variables. Other parameters were set under the following constraints: $P(F_1) = (1 - p_r)^N$, $p_r = p_v\, p_{occ}$, and $p_{occ} = k_1/(k_1 + b_1)$.

## Behavioral tests

### Trace fear conditioning

For tFC (*Gilmartin et al., 2013*), individual rats were placed in the training chamber ($30.5 \times 25.4 \times 30.5\ cm^3$; Coulbourn Instruments) consisting of metal grid floor connected to an electrical stimulator (H10-11R-TC-SF, Coulbourn Instruments) for delivery of a footshock. After a 5 min habituation period, rats received five pairings of a 20 s tone CS (2.5 kHz, 75 dB), followed by empty 20 s trace period and a US (1 s, 0.7 mA footshock) with a pseudorandom inter-trial interval (ITI) of $240 \pm 20$ s. During this training session, rats learned to associate the auditory CS with the shock US. The next day, the rats were tested for memory of CS-US trace association (called tone test). The acquisition of fear learning, often indicated by the extent of freezing to the CS, was assessed in a novel chamber (a white hexagonal enclosure) in a dimly lit room. During this tone (or retrieval) test, rats were first allowed to explore the new context for 5 min (habituation), followed by twelve 20 s CS presentations with a variable ITI of $120 \pm 20$ s. This entire session was repeated on the next day for assessment of extinction learning. Identities of the chambers were also determined with the presence of distinct odors; the training and testing chambers were cleaned with 70% ethanol and 1% acetic acid, respectively, before and after each session. For all experimental sessions, the activity of animals in the chambers was recorded at 30 frames per second using the EthoVision XT (Noldus Information Technology, Wageningen, Netherlands) and stored as a video file. The freezing behavior, defined as behavioral immobility other than respiratory movement, was assessed using an open-source video analysis pipeline ezTrack (*Pennington et al., 2019*). The percentage of freezing was calculated as a total duration of freezing divided by the total duration of observation.

### Elevated plus maze

EPM test was performed using a plus-shaped apparatus elevated 60 cm above the ground. The maze consisted of two open arms ($50 \times 10 \times 0.5\ cm^3$ (H)), two closed arms ($50 \times 10 \times 40\ cm^3$ (H)), and a center area ($10 \times 10\ cm^2$). Rats were individually placed on the center area, facing an open arm, and the path of the animal was recorded with a video camera for 5 min. We analyzed the total distance traveled for 5 min and time spent in each arm. Animals were tracked offline using the open-source tracking system ezTrack (*Pennington et al., 2019*).

### Open field test

Open field exploration test (OFT) was performed in an open field apparatus in a dimly lit room. The open field consisted of a white plastic board (1.20 m in diameter) surrounded by white plastic walls (50 cm in height). Individual rats were placed in a center zone (0.6 m in diameter) and allowed to freely explore the apparatus for 10 min. Time spent in the center zone and total distance traveled were analyzed for the initial 5 min. Animals were tracked offline using the open-source tracking system ezTrack (*Pennington et al., 2019*).

## Histology and immunohistochemistry

At the end of the behavioral experiments, rats were anesthetized using isoflurane inhalation and transcardially perfused with PBS followed by 4% paraformaldehyde (PFA, T&I, Korea). Brains were

extracted and post-fixed in 4% PFA solution for 24 hr. Perfused brains were sliced at 100 µm for estimation of GFP expression or 50 µm for immunohistochemistry. For analysis of c-Fos expression, rats were returned to their home cages immediately after the training session and perfused approximately 90 min later (*Arime and Akiyama, 2017*). Brains were then cryopreserved, frozen, and sectioned coronally. Immunofluorescence staining was conducted on free-floating slices using a rabbit monoclonal anti-c-fos (1:1000, Cell Signaling Technology, cat# 2250 s) and mouse monoclonal anti-GAD67 (1:500, Millipore, cat# mab5406). Fluorescent secondary antibodies included Cy5-conjugated goat anti-rabbit IgG (1:1000, Abcam, cat# ab97077) and Cy3-conjugated goat anti-mouse IgG (1:1000, Abcam, cat# ab97035). Brain slices were permeabilized and blocked with PBS containing 0.5% Triton X-100 and 5% normal goat serum for 1 hr at room temperature. Slices were incubated overnight at 4°C with primary antibodies, followed by 4 hr incubation with secondary antibodies at room temperature. Slices were then mounted with Antifading Mounting medium (Abcam, cat# ab104139). Confocal images were acquired using a Leica TCS SP8 microscope set to the same laser intensity and acquisition parameters to compare the immunosignals in sections from Scr and Syt7 KD groups. For cell counting and colocalization, fluorescent protein-positive cells were automatically quantified with spot-detection algorithms in Imaris 9.5 (Oxford instruments).

## Data analysis

Data were analyzed and presented using ClampFit (Molecular Devices), Igor Pro (WaveMetrics), MATLAB (Mathworks), and Prism (GraphPad). Data points and error bars represent the mean ± standard error of the mean. All statistical analyses were conducted using two-tailed comparisons. Sample sizes and statistical tests for each group are detailed in the figure legends.

## Numerical integration of the simple refilling model

This model posits reversible docking of vesicles at finite release sites ($N_{max}$) with forward ($k_1$) and reverse rate constants ($b_1$) (*Hosoi et al., 2007*; *Neher and Sakaba, 2008*).

$$\underset{b_1}{\overset{k_1}{\rightleftharpoons}} \, n \, \overset{p_v}{\longrightarrow}$$

According to this scheme, $dn/dt = k_1 u - b_1 n - p_v n \delta(t - t_{AP})$, where $u$ is the number of unoccupied sites (= $N_{max} - n$), $t_{AP}$ is the timing of AP firing, and $\delta$ is the Dirac delta function. The $p_v$ was assumed to be unity (*Figure 3—figure supplement 2*). The sum of basal $k_1$ (= $k_{1,b}$) and $b_1$ was estimated as 23/s from the dependence PPR on ISIs (*Figure 3—figure supplement 2Ab*). From the baseline occupancy of 0.3 and $p_{occ} = k_{1,b} / (k_{1,b} + b_1)$, we could estimate baseline $k_1$ and $b_1$ as 6.9/s and 16.1/s, respectively. $Ca^{2+}$ and Syt7-dependent increase of $k_1$ was modeled as $k_1 = k_{1,b} + K_{1,max} [Syt7: n \, Ca^{2+}]$, where $[Syt7: n \, Ca^{2+}]$ represents the fraction of full $Ca^{2+}$-bound Syt7. For deterministic simulations, we time-integrated a set of differential equations for each model using Euler methods with a time step of 1 ms. All calculations were performed on the platform of MATLAB (R2022b, Mathworks, USA).

## Acknowledgements

We thank Dr. E Neher and Dr. A Marty for critical reading of this manuscript and invaluable comments. Funding: This study was supported by grants from the National Research Foundation of Korea (RS-2024-333669 to S-H Lee; 2021R1I1A1A01059646 to Y Kim) and Seoul National University Hospital (2024).

## Additional information

### Funding

| Funder | Grant reference number | Author |
| --- | --- | --- |
| National Research Foundation of Korea | RS-2024- 333669 | Suk-Ho Lee |

| Funder | Grant reference number | Author |
|---|---|---|
| Seoul National University Hospital | | Suk-Ho Lee |
| National Research Foundation of Korea | 2021R1I1A1A01059646 | Yujin Kim |

The funders had no role in study design, data collection and interpretation, or the decision to submit the work for publication.

## Author contributions

Jiwoo Shin, Conceptualization, Data curation, Validation, Investigation, Visualization, Methodology, Writing – original draft, Writing – review and editing; Seung Yeon Lee, Investigation, Visualization, Methodology, Writing – review and editing; Yujin Kim, Formal analysis, Supervision, Funding acquisition, Validation, Investigation, Writing – review and editing; Suk-Ho Lee, Conceptualization, Data curation, Formal analysis, Supervision, Funding acquisition, Validation, Writing – original draft, Writing – review and editing

## Author ORCIDs

Jiwoo Shin ⓘ https://orcid.org/0009-0001-2580-0368
Seung Yeon Lee ⓘ https://orcid.org/0009-0009-5199-7369
Suk-Ho Lee ⓘ https://orcid.org/0000-0003-4117-5619

## Ethics

All animal procedures were approved and performed in accordance with the Institutional Animal Care and Use Committee at Seoul National University (SNU200522-1).

Reviewer #3 (Public review): https://doi.org/10.7554/eLife.102923.6.sa1
Author response https://doi.org/10.7554/eLife.102923.6.sa2

# Additional files

## Supplementary files

MDAR checklist

## Data availability

All data generated or analyzed during this study are included in the manuscript and supporting files. Source data files are provided in this paper. MATLAB codes for the analyses and simulation are presented in this article.

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
