## [Editor Report · eLife Assessment]

This **valuable** work explores how synaptic activity encodes information during memory tasks. All reviewers agree that the work is of very high quality and that the methodological approach is praiseworthy. The experimental data support the possibility that phospholipase diacylglycerol signaling and synaptotagmin 7 (Syt7) dynamically regulate the vesicle pool required for presynaptic release. Overall, this is a **convincing** study.

---

## [Referee Report · Reviewer #3 (Public review)]

The new results fill a key gap in the logic by strongly supporting the foundational premise that the very quickly reverting paired pulse depression at layer 2/3 synapses is caused by pool depletion. They are particularly critical because a previous study (Dobrunz, Huang, and Stevens, 1997) showed that a similar phenomenon is caused by a completely different category of mechanisms at Schaffer collateral synapses. This does not seem to be a case where the previous study was incorrect because, unlike here, synaptic strength at Schaffer collateral synapses is highly sensitive to extracellular Ca2+. Overall, such a fundamental difference between layer 2/3 and Schaffer synapses is highly noteworthy, given the similarities at the level of morphology and timing, and should be highlighted in the Discussion as an important result of its own. My only hesitation is that the authors do not seem to have done the control experiments, that I suggested, that would have confirmed that the synaptic strength remains stable when switching back to 1.3 mM Ca2+.

---

## [Author Response]

The following is the authors’ response to the previous reviews

**Public Reviews:**

**Reviewer #3 (Public review):**
To summarize: The authors' overfilling hypothesis depends crucially on the premise that the very quickly reverting paired-pulse depression seen after unusually short rest intervals of << 50 ms is caused by depletion of release sites whereas Dobrunz and Stevens (1997) concluded that the cause was some other mechanism that does not involve depletion on. The authors now include experiments where switching extracellular Ca2+ from 1.2 to 2.5 mM increases synaptic strength on average, but not by as much as at other synapse types. They contend that the result supports the depletion on hypothesis. I didn't agree because the model used to generate the hypothesis had no room for any increase at all, and because a more granular analysis revealed a mixed population with a subset where: (a) synaptic strength increased by as much as at standard synapses; and yet (b) the quickly reverting depression for the subset was the same as the overall population.The authors raise the possibility of additional experiments, and I do think this could clarify things if they pre-treat with EGTA as I recommended initially. They've already shown they can do this routinely, and it would allow them to elegantly distinguish between pv and pocc explanations for both the increases in synaptic strength and the decreases in the paired pulse ratio upon switching Ca2+ to 2.5 mM. Plus/minus EGTA pre-treatment trials could be interleaved and done blind with minimal additional effort.Showing reversibility would be a great addition too, because, in our experience, this does not always happen in whole-cell recordings in ex-vivo tissue even when electrical properties do not change. If the goal is to show that L2/3 synapses are less sensitive to changes in Ca2+ compared to other synapse types - which is interesting but a bit off point - then I would additionally include a positive control, done by the same person with the same equipment, at one of those other synapse types using the same kind of presynaptic stimulation (i.e. ChRs).Specific points (quotations are from the Authors' rebuttal)(1) Regarding the Author response image 1, I was instead suggesting a plot of PPR in 1.2 mM Ca2+ versus the relative increase in synaptic strength in 2.5 versus in 1.2 mM. This continues to seem relevant.

Complying with your suggestion, we studied the effects of external [Ca^2+^] ([Ca^2+^]_o_) after pre-incubating the slice in aCSF containing 50 μM EGTA-AM, and added the results as Figure 3—figure supplement 3C-D. Elevation of ([Ca^2+^]_o_) from 1.3 to 2.5 mM produced no significant change in either baseline EPSC amplitude or PPR, supporting that the p_v_ is already saturated at 1.3 mM [Ca^2+^]_o_ and implying that the modest Ca^2+^ dependence of baseline EPSCs and PPR in the absence of EGTA (Figure 3—figure supplement 3A-B) is mediated by the change in baseline vesicular occupancy of release sites (p_occ_) rather than fusion probability of docked vesicles (p_v_).

We found some correlation of high Ca^2+^-induced relative increase in synaptic strength with the PPR at low Ca^2+^ (Author response image 1-A). But this correlation was abolished by pre-incubating the slices in EGTA-AM too (Author response image 1-B). It should be noted that high PPR does not always mean low p_v_. For example, when the replenishment is equal between high and low baseline p_occ_ synapses, the PPR would be higher at low p_occ_ synapses than that at high p_occ_ synapses, even if p_v_ is close to unity. Therefore, high baseline release probability (Pr), whatever it is attributed to high p_v_ or high p_occ_, can result in low PPR, considering that Pr = p_occ_ x p_v_.

As we have already mentioned in our previous letter, the relationship of PPR with refilling rate is complicated and can be bidirectional, whereas an increase in p_v_ always results in a reduction of PPR. For example, PPR can be reduced by both a decrease and an increase in the refilling rate (Figure 2— figure supplement 1 and Lin et al., 2025). Therefore, the PPR analysis alone is insufficient to differentiate the contributions of p_v_ and p_occ_ Thanks to your suggestion, we could resolve this ambiguity by the EGTA-AM pre-incubation study (Figure 3—figure supplement 3C-D).

**Author response image 1. sa2fig1:** Plot of PPR at low [Ca^2+^]_o_ (1.3 mM) as a function of the baseline EPSC at high [Ca^2+^]_o_ (2.5 mM) normalized to that at low [Ca^2+^]_o_ measured at recurrent excitatory synapses in L2/3 of the prelimbic cortex under the conditions without EGTA-AM (A) and after pre-incubating the slices in EGTA-AM (50 μM) (B).

(2) "Could you explain in detail why two-fold increase implies pv < 0.2?"(a) start with power((2.5/(1 + (2.5/K1) + 1/2.97)),4) = 2^*^power((1.3/(1 + (1.3/K1) + 1/2.97)),4);(b) solve for K1 (this turns out to be 0.48);(c) then implement the premise that pv -> 1.0 when Ca2+ is high by calculating Max = power((C/(1 + (C/K1) + 1/2.97)),4) where C is [Ca] -> infinity.(d) pv when [Ca] = 1.3. mM must then be power((1.3/(1 + (1.3/K1) + 1/2.97)),4)/Max, which is <0.2. Note that modern updates of Dodge and Rahamimoff typically include a parameter that prevents pv from approaching 1.0; this is the gamma parameter in the versions from Neher group.

Thank you very much for your kind explanation. This interpretation, however, based on the premise that pv is not saturated at low[Ca^2+^]_o_, and that Pr = p_v_. In the present study, however, we presented multiple convergent lines of evidence supporting that p_v_ is already saturated at 1.3 mM [Ca^2+^]_o_ as follows: (1) little effect of EGTA-AM on the baseline EPSCs (Figure 2—figure supplement 1); (2) high double failure rates (Figure 3—figure supplement 2); (3) little effect of high [Ca^2+^]_o_ on baseline EPSC (Figure 3—figure supplement 3). Therefore, our results suggest that the classical Dodge-Rahamimoff fourth-power relationship can not be applied to estimate p_v_ at the L2/3 recurrent excitatory synapses.

(3) "If so, we can not understand why depletion-dependent PPD should lead to PPF." When PPD is caused by depletion and pv < 0.2, the number of occupied release sites should not be decreased by more than one-filth at the second stimulus so, without facilitation, PPR should be > 0.8. The EGTA results then indicate there should be strong facilitation, driving PPR to something like 1.2 with conservative assumptions. And yet, a value of < 0.4 is measured, which is a large miss.

As mentioned above, the framework used for inferring that p_v_ < 0.2, the Dodge-Rahamimoff equation, is not applicable to our experimental system. Consequently, the subsequent deduction— that depletion-dependent PPD should logically lead to PPF—is based on a model that does not compatible with aforementioned multiple convergent lines of evidence, which supports high p_v_ rather than the low p_v_ facilitation model.

(4) Despite the authors' suggestion to the contrary, I continue to think there is a substantial chance that Ca2+-channel inactivation is the mechanism underlying the very quickly reverting paired-pulse depression. However, this is only one example of a non-depletion mechanism among many, with the main point being that any non-depletion mechanism would undercut the reasoning for overfilling. And, this is what Dobrunz and Stevens claimed to show; that the mechanism - whatever it is - does not involve depletion. The most effective way to address this would be affirmative experiments showing that the quickly reverting depression is caused by depletion after all. Attempting to prove that Ca2+channel inactivation does not occur does not seem like a worthwhile strategy because it would not address the many other possibilities.

We have systematically ruled out alternative possibilities that may underlie the strong PPD observed at our synapses and demonstrated that it arises from high p_v_-induced vesicle depletion through multiple independent lines of evidence. First, we excluded (1) AMPAR desensitization or saturation (Figure 1—figure supplement 5), (2) Ca^2+^ channel inactivation (Figure 2—figure supplement 2), (3) channelrhodopsin inactivation (Figure 1—figure supplement 2), (4) artificial bouton stimulation (Figure 1—figure supplement 4), and (5) transient vesicle undocking (Figure 5; addressed in our previous rebuttal). Second, EGTA-AM experiments (Figure 2, Figure 2—figure supplement 1) revealed that release sites are tightly coupled to Ca^2+^ channels, and that EGTA further exacerbates PPD. Third, we validated high baseline p_v_ through analysis of double failure rates (Figure 3—figure supplement 2). Fourth, the minimal increase in baseline EPSCs upon elevation of external [Ca^2+^] (Figure 3—figure supplement 3) further supports that baseline p_v_ is already saturated at low [Ca^2+^]_o_. Additionally, to further validate our hypothesis, we performed the specific experiment suggested by the reviewer. We have now added EGTA pre-incubation experiments (Figure 3—figure supplement 3C-D) and have revised the manuscript. Specifically, when slices were pre-incubated with 50 μM EGTA-AM, elevation of extracellular [Ca^2+^] from 1.3 to 2.5 mM produced no significant change in either baseline EPSC amplitude or PPR, strongly supporting that the high [Ca^2+^]_o_ effects in the absence of EGTA are primarily mediated by changes in p_occ_ rather than p_v_

(5) True that Kusick et al. observed morphological re-docking, but then vesicles would have to re-prime and Mahfooz et al. (2016) showed that re-priming would have to be slower than 110 ms (at least during heavy use at calyx of Held).

As previously discussed, Kusick et al. (2020) demonstrated that the transient destabilization of the docked vesicle pool recovers very rapidly within 14 ms after stimulation. This implies that any posts stimulation undocking events are likely recovered before the 20 ms ISI used in our PPR experiments. Consequently, transient undocking/re-docking events are unlikely to significantly influence the PPR measured at this interval. Furthermore, regarding the slow re-priming kinetics (>100 ms) reported by Mahfooz et al. (2016) and Kusick et al., (2020), our 20 ms ISI effectively falls into a me window that avoids the potential confounds of both processes: it is long enough for the rapid morphological recovery (~14 ms) of docked vesicles to occur, yet too short for the slow re-priming process to make a substantial contribution. Furthermore, Vevea et al. (2021) showed that post-stimulus undocking is facilitated in synaptotagmin-7 (Syt7) knockout synapses. In our study, however, Syt7 knockdown did not affect PPR at 20 ms ISI, suggesting that the undocking process described in Kusick et al. (2020) is not a major contributor to the PPD observed at 20 ms intervals in our experiments. Therefore, we conclude that the 20 ms ISI used in our experiments falls within a me window that is influenced neither by the rapid undocking (<14 ms) reported nor by the slow re-priming process (>100 ms).